**Centennial to millennial climate variability in the far northwestern Pacific (off Kamchatka)**
**and its linkage to the East Asian monsoon and North Atlantic from the Last Glacial**
**Maximum to the Early Holocene**
Sergey A. Gorbarenko [1], Xuefa Shi [2], Min-Te Chen [3], Galina Yu. Malakhova[4],
Aleksandr A. Bosin [1], Yanguang, Liu [2], Jianjun, Zou [2]
[1] V.I. Il'ichev Pacific Oceanological Institute, Russia
[2] First Institute of Oceanography, SOA, China
[3] National Taiwan Ocean University
[4] North-East Interdisciplinary Science Research Institute FEB RAS, Russia
**Abstract**
High resolution reconstructions based on productivity proxies and magnetic properties
measured in sediment core 41-2 (off Kamchatka), reveal prevailing centennial-millennial
productivity/climate variability in the northwestern (NW) Pacific from the Last Glacial
Maximum (LGM) to the Early Holocene (EH). The age model of core 41-2 is established by
Accelerator mass spectrometry (AMS) [14]C dating using foraminifera shells, and by the
correlation of the productivity cycles and relative paleomagnetic intensity records with the same
cycles and records of the well-dated nearby core SO201-12KL. Our results show that a
pronounced feature of centennial-millennial productivity/climate cycles in the NW Pacific
occurred synchronously with the summer East Asian Monsoon (EAM) at sub-interstadial scale
during the LGM (3 cycles), Heinrich Event 1(3 cycles), Bølling/Allerød warming (4 cycles), and
over the EH (3 cycles). Comparison of the centennial-millennial NW Pacific
productivity/climate cycles with the variability of the Antarctic temperature of the EPICA

Dronning Maud Land (EDML) ice core suggests a "push" effect of Southern hemisphere temperature gradients on the intensifications of the summer EAM. Besides the linkages of NW Pacific high productivity and the summer EAM, we observed that five low productivity cycles during the EH are nearly synchronous with cooling in Greenland, with weakening of the summer EAM, and with decreases in solar irradiance. We propose that such centennial-millennial productivity/climate variability in the NW Pacific, associated with sub-interstadials/stadials in the EAM from the LGM to the EH, is a persistent regional feature and is quasi-synchronous with the Greenland/North Atlantic short-term changes. We speculate that such climate variability was also forced by changes in the Atlantic meridional overturning circulation, coupled with the Intertropical Convergence Zone shifting, and reorganization of the northern westerly jets.

**1. Introduction**

Model simulations and proxy-based interpretations have led to contradictory results concerning the millennial environmental variability in the northwestern (NW) Pacific, and its underlying mechanisms during the last deglaciation. These model and proxy studies have suggested either in-phase relationships of deglacial variability between the North (N) Atlantic and N Pacific (Caissie et al., 2010; Chikamoto et al., 2012; Kienast and McKay, 2001; Seki et al., 2002) or out-of-phase responses (Gebhardt et al., 2008; Okazaki et al., 2010; Sarnthein et al., 2006). The in-phase relationship has been attributed to rapid atmospheric teleconnections in the N hemisphere on a decadal time scale (Max et al., 2012). The winter Arctic Oscillation, which resembles the North Atlantic Oscillation, directly influences the surface air temperature and sea level pressure over the region northwards of 35ºN in East Asia. In turn, the Siberian High significantly influences the East Asian Winter Monsoon (Wu and Wang, 2002). The out-of-phase response, however, was proposed to be driven by a seesaw mechanism, with oceanic readjustments between the Atlantic meridional overturning circulation (AMOC) and the Pacific meridional overturning circulation (Saenko et al., 2004). Recent studies on high-resolution and precisely-dated sediment cores from the subarctic NW Pacific, the Sea of Okhotsk, and the western Bering

Sea show a deglacial sea surface temperature (SST) evolution similar to the northeastern (NE)
Pacific, and to the N Atlantic and Greenland temperature variability (Max et al., 2012). These
studies suggest a close link to deglacial variations in the AMOC, associated with rapid
atmospheric teleconnections, which were responsible for a quasi-synchronous SST development
between the N Atlantic and N Pacific during the last deglaciation. On the basis of high resolution
X-ray fluorescence (XRF) and sediment color reflectance studies of western Bering Sea cores,
Riethdorf et al. (2013) further suggest a close link between millennial-scale productivity changes
and the Dansgaard-Oeschger variability registered in the North Greenland Ice Core Project
(NGRIP) ice core, which had been interpreted as supporting the atmospheric coupling
mechanism. A study comparing the subarctic N Pacific dust record to dust content in the NGRIP
ice core also shows synchronicity of the timing of abrupt millennial changes during the last 27 ka
(Serno et al., 2015). Furthermore, a recent study by Praetorius and Mix (2014), based on
multidecadal-resolution foraminiferal oxygen isotope records from the Gulf of Alaska, reveals a
synchronicity of rapid climate shifts between the N Atlantic/Greenland (NGRIP core record) and
the NE Pacific between 15.5 and 11 ka. During the Holocene and Heinrich Event (HE) 1, inverse
relationships between the Atlantic/Pacific are suggested in this paper, while the short-term
variability is either not sufficiently resolved or is decoupled.
All of these instances indicate that a lack of high resolution proxy records in the NW
Pacific prohibits precise assessments of any possible climatic teleconnection mechanisms across
the basins. Although abrupt centennial-millennial precipitation anomalies from the Last Glacial
Maximum (LGM) to the Holocene have been reported in cave sediment $\delta^{18}O$ records of the East
Asian monsoon (EAM) (Dykoski et al., 2005; Wang et al., 2001, 2005, 2008; Yuan et al., 2004),
the timing and trend of variability of Early Holocene (EH) regional climate changes are still
controversial. In particular, though the EH climate started with a strong warming in most cases, a
Hani peat $\delta^{18}O$ record from northeastern China instead indicates cooling events which are
primarily superimposed on a Holocene long-term warming trend (Hong et al., 2009).

Here the high resolution results of a suite of productivity proxies, magnetic properties, and

lithological changes from the NW Pacific sediment core LV 63-41-2 (hereafter, 41-2) (off

Kamchatka) are presented and reveal a sequence of centennial-millennial climate/productivity

variability from 20 ka to 8 ka. An age model of this core was constructed using Accelerator mass

spectrometry (AMS) [14]C dating and by correlating the productivity cycles and relative

paleomagnetic intensity (RPI) variability with those of the well-dated nearby core SO-201-12KL

(hereafter, 12KL) (Max et al., 2012, 2014). Using methodologically robust age controls, it is

possible to infer a tight linkage between the centennial-millennial productivity variability in the

NW Pacific, and the sub-interstadial summer EAM intensifications expressed in cave sediment

$\delta^{18}O$ records. These results enable the further investigation of any mechanisms controlling the in-

phase relationships of the centennial-millennial variability in the NW Pacific/EAM and those

underlying the Greenland/N Atlantic and Antarctic climate changes during the LGM – HE 1 –

Bølling/Allerød (B/A) – Younger Dryas (YD) – EH (~20–8 ka).

**2 Materials and methods**

**2.1 Coarse fraction measurement**

Sediment core 41-2 was recovered in the NW Pacific off the Kamchatka Peninsula (water

depth 1924 m; 52°34' N, 160°06' E; core length 467 cm) during the joint Russian-Chinese

expedition at R/V "Akademik M.A. Lavrentyev" in 2013. The weight percentage of the coarse

fraction (CF) >63 μm and <2000 μm, sampled every 1 cm and separated by sieve washing, was

calculated as a ratio of the CF weight to the weight of the dry bulk sediment. Semi-quantitative

estimates were made of the amount of various components in the sediment CF, including

terrigenous and volcanogenous particles (tephra), benthic and planktonic foraminifera, diatom

frustules, and radiolarians, using a microscope to roughly estimate the proportions of different

components in the sediment (Rothwell, 1989). The indicators of materials mainly transported to

the study region by sea ice, such as CF and MS of sediments (Gorbarenko et al., 2003; Lisitzin,

2002; Sakamoto et al., 2005), are used as an ice rafted debris (IRD) proxy. Semi-quantitative

estimates of the amount of terrigenous and volcanic particles in sediment CF allow the
determination of core intervals with insignificant amounts of tephra, and therefore intervals with
implications for CF and MS as an IRD index.
**2.2 Chlorin content measurement**
Chlorin content is assumed to reflect changes in primary surface ocean productivity,
because continental-derived chlorophyll does not contribute to the chlorin content in deep marine
sediment (Harris et al., 1996). The chlorin content in core 41-2 was measured at 1 cm resolution,
and at 2 cm resolution in core 12KL through the whole core, as in Harris et al. (1996), modified
using a Shimadzu UV-1650PC spectrophotometer (Zakharkov et al., 2007).
**2.3 Total organic carbon (TOC), calcium carbonate, and color b\* measurements**
Contents of TOC, $CaCO_3$, and biogenic opal in deep sea sediments are usually used as
key parameters to assess paleoproductivity (Berger et al., 1989; Narita et al., 2002; Prahl et al.,
1989; Seki et al., 2004). Shipboard color b\* values correlate well with the changes in biogenic
opal content in sediment cores (Nürnberg and Tiedemann, 2004) and are widely used as a
paleoproductivity proxy in the NW Pacific and its marginal seas (Gorbarenko et al., 2012; Max
et al., 2012; Riethdorf et al., 2013).
The total carbon content and inorganic carbon in core 41-2 were measured every 2 cm
throughout the core depth by coulometry using an AN-7529 analyzer (Gorbarenko et al., 1998).
TOC content was determined by calculating the difference between total carbon and inorganic
carbon content. A color b\* index (psychometric yellow-blue chromaticness) was measured with
1 cm resolution using a Minolta CM-2002 color reflectance spectrophotometer (Harada, 2006).
**2.4 Radiocarbon dating (AMS [14]C)**
AMS [14]C-ages were measured in monospecific samples of the planktic foraminifera
*Neogloboquadrina pachyderma* sinistral (*N. pachyderma* sin.) from the 125–250 μm fraction,
and benthic foraminifera *Epistominella pacifica* and *Uvigerina parvocostata* from the 250–350
μm fraction of the core. The radiocarbon dating was performed by Dr. John Southon at the Keck
Carbon Cycle AMS Facility (UCIAMS) in the Earth System Science Department of the
University of California, USA. The constant reservoir age (900 ± 250 yr) of the NW Pacific
surface water (Max et al., 2012) was adopted in this study to convert the $^{14}C$ ages of the samples
into calendar ages, in order to establish consistent AMS $^{14}C$ chronologies of cores 41-2 and
12KL. All reservoir age-corrected $^{14}C$ data were converted to calendar age by using Calib Rev
6.0 (Stuiver and Reimer, 1993) with the Marine13 calibration curve (Reimer et al., 2013).When
using benthic foraminifera for AMS $^{14}C$ dating on the cores, an age difference of 1400 yrs is
taken between coexisting benthic and planktic foraminifera ages (Max et al., 2014).
**2.5 Magnetic property measurements**
Magnetic properties were measured at 2.2 cm resolution in both cores. The volume
magnetic susceptibility (MS) of these samples was measured using an AGICO MFK1-FA
device. The characteristic remnant magnetization (ChRM) of the samples was measured in the
same way, by studying the stability of natural remanent magnetization (NRM) in the alternative
magnetic fields of up to 80–100 mT, on the basis of analysis of Zijderveld vector plots, using an
AGICO LDA-3A device and rock-generator AGICO JR-5a (Zijderveld, 1964). The module and
direction of NRM were measured on a JR-5A rock-generator after the stepwise demagnetization
of reference samples by alternating magnetic fields with a vanishing amplitude (Malakhov et al.,
2009). Ahysteretic remanent magnetization (ARM) was generated using an AGICO AMU-1A
device and measured using the JR-5A rock-generator. The relative paleomagnetic intensity (RPI)
of the studied core sediments was determined by the normalization of the ChRM after
demagnetization at 20 mT by ARM (ChRM/ARM) (Tauxe, 1993). The sediment paramagnetic
magnetization (PM) was measured for each sample from curves of magnetic hysteresis using a J
Meter coercive spectrometer at Kazan State University, Kazan, Russia (Enkin et al., 2007;
Jasonov et al., 1998).
Past relative paleomagnetic intensity (RPI) value changes in response to variability in the
Earth's magnetic field present an independent chronological instrument of marine and
continental sediments (Channell et al., 2009), and are widely used for sediment correlation and
chronology (Kiefer et al., 2001; Riethdorf et al., 2013). PM was formed in marine sediments of
silicate, paramagnetic iron sulphide (FeS), and fine clay minerals transported from land as an
eolian dust through atmospheric circulation by westerly jets. Therefore, the sediment PM may
serve as a proxy of the land aridity and/or atmospheric circulation pattern changes in response to
climate change. MS was mainly formed by ferromagnetic minerals delivered together with
terrigenous materials from adjacent land, and is therefore related to IRD. It is the main transport
agent of clastic material input into the sediment of the NW Pacific and its marginal seas
(Gorbarenko et al., 2003; Lisitzin, 2002; Sakamoto et al., 2005).
**2.6 XRF measurements**
The elemental composition of core 41-2, given in peak area (counts per second, cps), was
measured at 0.5 cm resolution using the Itrax XRF core scanner at the First Institute of
Oceanography, State Oceanic Administration, China. The Itrax XRF core scanner was set to 20 s
count times, 30 kV X-ray voltage, and an X-ray current of 20 mA. Though absolute elemental
concentrations are not directly available from the micro-XRF measurements, the count values
can be used as estimates of the relative concentrations. The count values may be influenced by
changes in the physical properties of the sediment, such as the surface roughness of the core
(Röhl and Abrams, 2000). However, the grain size of the 41-2 core is rather fine, and the surface
has been processed to be as flat as possible to minimize any effects due to changing physical
properties or roughness during the scanning.
In this study, attention was paid to the scanning results for estimating biogenic Ba, Br, and
Si (Ba-bio, Br-bio and Si-bio respectively) contents in the sediment cores, which serve as proxies
for productivity. The content of Ba-bio was estimated through the subtraction of its terrigenous
component (Ba-ter) from the total bulk Ba concentration in the sediment (Ba-tot). The
terrigenous component, in turn, was calculated from empirical regional $(Ba/Al)_{ter}$ ratios in the
sediment core with the lowest Ba-tot contents:

Ba-bio = Ba-tot – $(Ba/Al)_{ter}$*Al (Goldberg et al., 2005).

Br-bio and Si-bio were calculated using the same technique.
Earlier it was shown that the non-destructive, high resolution X-ray Fluorescence (XRF)
measurements of Ba-bio, Br-bio, and Si-bio by a core scanner or synchrotron radiation are
consistent with analytically measured Ba-bio, TOC, and biogenic opal, respectively, and
therefore may be used as paleoproductivity proxies (Goldberg et al., 2005; Nürnberg and
Tiedemann, 2004; Riethdorf et al., 2016). Ba-bio is formed during the decay of organic matter in
the water column and the uptake of Ba in settling particles (Dymond et al., 1992), and has been
previously used as a paleoproxy (Goldberg and Arrhenius, 1958; McManus et al., 1998). Si-bio,
related with biogenic opal in deep sea sediments, is usually used as a key parameter to assess
paleoproductivity (Berger et al., 1989; Narita et al., 2002; Seki et al., 2004). Br-bio content
measured using a core scanner is strongly correlated with TOC variability (Riethdorf et al.,
2016) and therefore may also be used as a paleoproductivity proxy.

**3. Results**

AMS radiocarbon data for core 41-2 are presented in Table 1. The variability of a suite of

productivity proxies (color b* and contents of TOC, chlorin, $CaCO_3$, Ba-bio, Si-bio, and Br-bio),
plus magnetic properties (RPI, sediment PM, and MS), are presented for core 41-2 versus depth
(Fig. 2). Increased productivity at the interval ~315–230 cm, according to several productivity
proxies and available AMS [14]C data, could be chronologically assigned to the B/A warming right
after the termination of the last glaciation (467–315 cm) (Fig. 2). The high productivity during
the B/A warming is a common feature in the far NW Pacific and its marginal seas (Galbraith et
al., 2007; Gorbarenko, 1996; Gorbarenko et al., 2005; Gorbarenko and Goldberg, 2005;
Keigwin, 1998; Seki et al., 2004). The interval at ~230–190 cm with a decreased trend of
productivity is likely associated with the YD cooling. After this low productivity/cold climate
event the high productivity/warm trend in the upper 190 cm of the core is presumably related to
the Holocene warming.
In core 41-2, the time resolutions of measured color b*, and chlorin, TOC, $CaCO_3$ content
and magnetic parameters (PM, MS, and RPI); and Ba-bio, Br-bio, and Si-bio concentrations over
the LGM-YD periods are nearly 30 years, 15 years, and 60 years respectively. The resolution is
high enough to allow the detection of centennial-millennial scale climate variability in the far
NW Pacific. Graphic correlation of the productivity proxies (chlorin, TOC, $CaCO_3$, Ba-bio, Br-
bio, Si-bio content, and color b*) and the PM record reveal quasi-synchronous centennial-
millennial productivity cycles likely associated with abrupt environmental variability (Fig. 2) via
mechanisms similar to previously established regularities at the orbital-millennial scale
(Broecker, 1994; Ganopolski and Rahmstorf, 2002; Sun et al., 2012). Therefore, it is suggested
that the sharp increase in productivity demonstrates the fast response of the NW Pacific
environment associated with abrupt regional warming, and vice versa, similar to interstadial
events in the NW Pacific and the Okhotsk and Bering Seas. The rises in temperature of surface
water and environmental amelioration in the NW Pacific, the Okhotsk and Bering Seas, and the
Sea of Japan are well correlated with interstadials in $\delta^{18}O$ records in the NGRIP ice core (North
Greenland Ice Core Project members, 2004) and in the Chinese cave stalagmites promoting to
increase in productivity at the millennial scale (Gorbarenko et al., 2005; Nagashima et al., 2011;
Schlung et al., 2013; Seki et al., 2002, 2004).
Each productivity proxy used here has its own specific limitations and peculiarities in its
response to environmental and primary productivity changes. For example, although
carbonaceous fossils (planktonic foraminifera and coccolithophorids) rain from the euphotic
layer derived by primary production, and provide the main carbonate input into the sediment,
$CaCO_3$ content in the deep sea sediment is mostly governed by climatically forced variability in
the deep water chemistry and carbonate ion concentration ($CO_3^{2-}$), resulting in different
carbonate preservation in the past (Yu et al., 2013). As for the Ba-bio proxy, Jaccard et al. (2010)
suggest that in the highly productive areas, barite dissolution has been observed under suboxic
conditions, precluding its application as a quantitative proxy to reconstruct past changes in
export production. Although it has been suggested that biogenic opal and TOC content, being
responsible for the accumulation of siliceous fossils, and siliceous plus carbonaceous fossils,
respectively, present basic key proxies for the assessment of productivity changes (Berger et al.,
1989), they vary in different ways at various times in sediments of the NW Pacific and its
marginal seas. For example, in the Okhotsk Sea biogenic opal content lags significantly relative
to TOC changes during the last deglaciation—the Late Holocene (Gorbarenko et al., 1998; Seki
et al., 2004). TOC content in the hemipelagic sediment includes the organic carbon formed by
marine primary production, and the terrigenous organic material delivered from land. The input
of which depends on the river runoff and sea level changes. Therefore, centennial-millennial
changes in different productivity proxies vary not exactly synchronously, depending on organic
matter transformation into a different proxy, and its subsequent preservation in the sediments.
The presentation of a wide range of productivity proxies allows different aspects of the
transformation of primary produced organic matter into different proxies, and their preservation
in sediment, to be considered. This approach provides a more reliable pattern of productivity
changes. Beside productivity proxies the PM record is also used, because the sediment PM
reflects the changes in the transportation of dust from continents by atmospheric circulation
associated with climate change. For the statistical assessment of the centennial-millennial
productivity variability, the productivity stack is calculated. It is an average of the normalized
data of each proxy, given equal weight (Fig. 3).
A graphic correlation of all the applied productivity proxies with the sediment
paramagnetic magnetization (PM) record shows that six short increased productivity/warmer
events happened during the last glacial, and four occurred during the B/A warming (Fig. 2, Table
2). During the EH five short lower productivity/colder events and three higher
productivity/warmer events were found. It is noted that a colder event at depth 117–122 cm with
an age of ~9.12 ka (Table 1) is well-correlated with the 9.3 ka cold event in the Greenland ice
core records (Rasmussen et al., 2014). Moreover, a colder event identified at depth 106–109 cm
in core 41-2 also links well with the 8.2 ka cold event in the Greenland ice cores, a well-known
chronostratigraphic marker in the Early to Middle Holocene boundary (Walker et al., 2012).
The share of tephra in the sediment CF shows relative low values below 130 cm, and
significantly increases in the upper part of the core (Fig. 2). Therefore, CF and MS records,
controlled by the tephra share in CF, indicate high IRD inputs in the sediment of the lower part
of the core, and a strong decrease towards the top in the interval 325–315 cm. MS and CF
records also show some increase of IRD input in the interval 230–200 cm, related to the YD
(Fig. 2).
The relative paleointensity (RPI), color b* records, and productivity stack of core 41-2
were compared with the RPI, PM, and several productivity proxies of nearby core 12KL versus
core depth (Fig. 3). The color b* index and Ca (analog of $CaCO_3$ content) of core 12KL were
obtained from Max et al. (2012, 2014) and PANGAEA Data Publisher for Earth and
Environmental Science (https://doi.pangaea.de/10.1594/PANGAEA.786201). The centennial-
millennial events with increased productivity shown in Fig. 2 were confirmed by the productivity
stack changes for core 41-2, and correlate well with productivity events for core 12KL outlined
by the productivity proxies and PM record; their correlation is also consistent with RPI
variability in both cores.
**4. Age model**
An age model of core 41-2 was constructed using all available AMS [14]C dating, with more
age control points identified by correlating the centennial-millennial events of the productivity
proxies, RPI, and PM of the studied core with those of the well-dated nearby core 12KL (Max et
al., 2012, 2014) (Fig. 3). The age tuning used in this study assumes a synchronous pattern of
productivity, RPI, and PM variability in the far NW Pacific since the last glacial, especially for
closely-located cores. With this conception of age model developments, the centennial-
millennial variability of productivity proxies with increased productivity events, relative
paleointensity (RPI) of Earth's magnetic field, and paramagnetic magnetization (PM) identified
in cores 41-2 and 12KL have to be closely matched in both cores over the last glaciation—the
B/A warming to the EH (Fig. 3). It was noted that the available model for core 12KL—the
Tiedemann/Max age model 2 (Max et al., 2012, 2014)—was based on AMS $^{14}$C data and the
correlation of the color b* index with the NGRIP $\delta^{18}$O curve (PANGAEA Data Publisher). By
adopting an age model of core 41-2, the AMS $^{14}$C dating of core 12KL of Max et al. (2012,
2014) was successfully projected to core 41-2 according to the correlation of related increased
productivity events and RPI values (Fig. 3). The color b* minimum in core 12KL at a depth of
706 cm, which Tiedemann and Max (PANGAEA Data Publisher) correlate with a minimum in
the NGRIP $\delta^{18}$O curve at 16.16 ka, is also clearly correlated with the color b* minimum of core
41-2 at a depth of 348 cm (Fig. 3). All correlated AMS $^{14}$C key points are also well-matched
with the measured RPI curves of both cores (Fig. 3). Core 41-2 AMS $^{14}$C data of 9.45 ka, 10.6
ka, 14.39 ka, and 14.61 ka at depths of 127.5 cm, 156 cm, 298 cm, and 306 cm, respectively, are
fairly close to the nearby projected $^{14}$C datum from core 12 KL (Table 3), and confirm the
validity of this age projection. Here the use of the $^{14}$C data of core 12KL is preferred, because
this core has a higher sedimentation rate, and planktonic foraminifera for these measurements
were picked from intervals with the highest Ca content, to significantly decrease a bioturbation
effect.

A close time correlation of these NW Pacific productivity increasing/environmental

amelioration events with sub-interstadials in the summer EAM becomes apparent after placing
the radiocarbon datum of both cores on the absolute U-Th dated $\delta^{18}$O record of Chinese cave
stalagmites (Wang et al., 2008) over the 20–8 ka (Fig. 3). Such inferred synchronicity of abrupt
NE Pacific productivity events and EAM sub-interstadials was used for further age model
construction. This was achieved by fine-tuning the increased productivity events with related
sub-interstadials of $\delta^{18}O$ Chinese stalagmites for a depth beyond the projected AMS $^{14}C$ data
(Fig. 3; Table 3).
**5. Discussion**
Within the constructed age model of core 41-2, different productivity proxies and magnetic
results were combined with similar data from core 12KL (Max et al., 2012, 2014). These data
reveal a sequence of noticeable centennial-millennial scale productivity cycles in the far NW
Pacific, which occurred in-phase with Chinese sub-interstadials (CsI) associated with a stronger
summer EAM (Wang et al., 2008) over the period 21–8 ka (Fig. 4). These linkages suggest the
centennial-millennial increased productivity events in the far NW Pacific were likely associated
with shifts to a warmer climate and/or higher nutrient conditions in surface water synchronously
with CsI of the summer EAM. High resolution records presented here show clearly that three
centennial-millennial increased productivity/environment amelioration events correlated with
CsI had occurred during the LGM, three CsIs during the HE 1, four CsIs during the B/A
warming, and three CsIs during the EH (Fig. 4; Table 2). The possible mechanisms responsible
for the in-phase relationships or the synchronicity of the centennial-millennial scale events
between the NW Pacific productivity and summer EAM are proposed and discussed below.
**5.1. N-S hemisphere climatic linkages of centennial-millennial climate/environment**
**changes over the LGM-HE 1-B/A warming**
The identification of any linkages between centennial-millennial climate changes in the
Northern Hemisphere (NW Pacific, EAM, and N Atlantic/Greenland) and the climate changes
recorded in Antarctic ice cores representative of the Southern Hemisphere is important for
deepening understanding of the mechanisms responsible for the timing and spatial propagation
patterns that resulted from abrupt variability in the global climate and environmental system. In
order to test these linkages, the centennial-millennial productivity/climate events in the NW
Pacific outlined by the productivity stack are correlated with a variety of other records. These
are: the highly resolved U-Th dated $\delta^{18}O$ records of the composite Hulu and Dongge stalagmites
(Dykoski et al., 2005; Wang et al., 2008); the ~20-year averaged resolution $\delta^{18}O$ and $Ca^{2+}$
content records of the GISP2 and NGRIP, with a five-point running mean on the annual-layer
counted GICC05 age scale (Rasmussen et al., 2014); the $\delta^{18}O$ record of the EPICA Dronning
Maud Land (EDML) ice core from Antarctica (EPICA Community Members, 2006) on the
methane synchronized timescale with the NGRIP core; and the Siberian climate calculated from
pollen records of the Lake Baikal region (Bezrukova et al., 2011) over the past 25 ka (Fig. 5).
The $Ca^{2+}$ content in the Greenland ice cores serves as a proxy for dust mobilization on the land,
and for transfer in the high latitudes of the N Hemisphere by an atmosphere governed by climate
and atmospheric circulation changes (Sun et al., 2012). It has been suggested that the nearly
synchronous ice core $\delta^{18}O$, and $Ca^{2+}$ millennial-scale changes reflect the shifting of the
Greenland atmospheric dust loading, which is closely linked with the atmospheric circulation
and climate changes in the high latitudes of the N Hemisphere, where the EAM plays an
important role (Ruth et al., 2007). Initially, the persistent millennial-scale changes shown in the
Greenland ice core records were defined as interstadials (GI) and stadials (GS) (Johnsen et al.,
1992), but have been refined by INTIMATE stratigraphy studies which introduced the
subdivision of the GI-1 into sub-interstadials GI-1a to GI-1e. Furthermore, the GS-2.1 was
subdivided into sub-stadials GS-2.1a (over the HE 1), GS-2.1b (LGM), and GS-2.1c (Björck et
al., 1998; Rasmussen et al., 2014) (Fig. 5).

During the construction of the age model, a strong correlation was established between the

centennial-millennial productivity/environment events in the NW Pacific cores, and the sub-
interstadials of the summer EAM over the LGM-HE 1-B/A (Fig. 5), suggesting a strong, causal
teleconnection. This suggests that, in addition to the six centennial-millennial
productivity/environment cycles over the LGM-HE 1 established in the NW Pacific cores,
another three abrupt events likely took place in the NW Pacific coeval with CsIs outlined by the
$\delta^{18}O$ of Chinese stalagmites over the interval 25–20 ka (Fig. 5). Therefore, it was found that
three EAM/NW Pacific sub-interstadials occurred within GS-2.1a (namely CsI-GS2.1-1, CsI-
GS2.1-2, and CsI-GS2.1-3), four CsIs occurred within GS-2.1b (CsI-GS2.1-4 to CsI-GS2.1-7),
and two occurred within GS-2.1c (CsI-GS2.1-8 and CsI-GS2.1-9) (Fig. 5).
It also has been noted that there are some $\delta^{18}O$ differences between coeval $\delta^{18}O$ values in
the Summit and NGRIP ice cores over the LGM-HE 1 period, which were likely governed by
changes in the N American Ice Sheet volume and N Atlantic sea-ice extent, resulting in changes
of the meridional gradients in the $\delta^{18}O$ of Greenland ice (Seierstad et al., 2014). Such differences
in the Summit/NGRIP $\delta^{18}O$ values may explain why the correlation of the EAM/NW Pacific
sub-interstadials with the Greenland sub-interstadials recorded in the $\delta^{18}O$ and $Ca^{2+}$ records of
the GISP2 and NGRIP cores was more clear during LGM, and less pronounced over the HE 1
(GS-2.1a) (Fig. 5).
On the basis of the high-resolution NGRIP core investigation (less than one year) over 15–
11 ka, Steffensen et al. (2008) have suggested that at the beginning of the GI, the initial northern
shift of the Intertropical Convergence Zone (ITCZ), identified from a sharp decrease of dust
within a 1–3 year interval, triggered an abrupt shift in Northern Hemisphere atmospheric
circulation. Such circulation pattern changes forced a more gradual change (over 50 years) of the
Greenland air temperature, associated with the reorganization of high latitude atmospheric
circulation and westerly jets. Evidence from a loess grain size record in the NW Chinese Loess
Plateau (Sun et al., 2012), implies a link between the changes in EAM strength and the
Greenland air temperatures over the past 60 ka, and suggests that a common force was driving
both changes (Sun et al., 2012). Using a coupled climate model simulation Sun et al. (2011)
investigated the effect of a slow-down of AMOC on the monsoon system, and found that a
stronger winter EAM, accompanied with a reduction in summer monsoon precipitation over East
Asia, supplies more dust to the Chinese Loess Plateau and likely also to the NW Pacific. This
study indicates that the AMOC is a driver of abrupt change in the EAM system, with the
northern westerlies as the transmitting mechanism from the N Atlantic to the Asian monsoon
regions. Other evidence of teleconnections between the EAM and N Atlantic on a millennial
timescale come from the investigation of sediment cores from the Sea of Japan. Nagashima et al.
(2011) infer that temporal changes in the provenance of eolian dust in sediments from the Sea of
Japan reflect changes in the westerly jet path over East Asia, which happened in-phase with the
Dansgaard-Oeschger cycles.

EPICA community members (2006) show that methane synchronization of the EDML and

the NGRIP $\delta^{18}$O records reveal one-to-one alignment of each Antarctic warming with a
corresponding stadial in the Greenland ice cores, implying a bipolar seesaw mechanism on these
time scales. Changes in the heat and freshwater flux were connected to the AMOC, and a
stronger AMOC leads to the increased transport of heat from the Southern Ocean heat reservoir.
As a result of EAM investigations Wang et al. (2001) have suggested that between 11,000 and
30,000 yr BP the Chinese interstadials (CI) recorded in $\delta^{18}$O calcite of cave stalagmites had
happened apparently synchronously with the GIs. Therefore, CIs were also likely related to
Antarctic cold events. In confirmation, smoothed warmer conditions in the Antarctic at 23.6–
24.3 ka were synchronous with abrupt climate cooling and increases in dust content in the
Greenland ice cores NGRIP and GISP2, coeval to HE 2 of the N Atlantic, and in-phase with the
weakening of the summer EAM (GS/CS-3.1) (Fig. 5). The Antarctic cooling since 23.4 ka was
accompanied by warming in Greenland, with two sharp interstadials GI-2.2 and GI-2.1
(Rasmussen et al., 2014) and China interstadial CI-2 coeval with sub-interstadial CsI-GS2.1-9
associated with summer EAM intensification (Fig. 5). Over the LGM period, most of the sub-
interstadials in the NW Pacific/summer EAM had occurred during abrupt Antarctic temperature
decreases, while during HE 1 sub-interstadial linkages between the N and S hemispheres are less
evident (Fig. 5).

It has also been suggested that a monsoon intensity index including the EAM was

controlled not only by Northern Hemisphere temperature ("pull" on the monsoon, which is more
intense during boreal warm periods), but also by the pole-to-equator temperature gradient in the
Southern Hemisphere ("push" on the monsoon, which is more intense during the boreal cold
periods) that leads to enhanced boreal summer monsoon intensity and its northward propagation
(Rohling et al., 2009; Rossignol-Strick, 1985; Xue et al., 2004). Since the summer EAM
transports heat and moisture from the West Pacific Warm Pool (WPWP) across the equator and
to higher northern latitudes (Wang et al., 2001), the temperature gradient in the Southern
Hemisphere "pushes" the summer EAM intensity by means of its influence on the
latitudinal/longitudinal migrations or expansion/contraction of the WPWP. This also explains the
difference in responses of the EAM and Greenland interstadials and sub-interstadials, because
the migration of the WPWP may have occurred more slowly than the atmospheric changes. The
changes in the $\delta^{18}O$ of Chinese cave stalagmites were more gradual then in the $\delta^{18}O$ of
Greenland ice cores, and were more similar to the Antarctic air temperature changes (Fig. 5).
During B/A warming when Antarctic temperatures decreased, four EAM sub-interstadials
(CsI-GI1-a to CsI-GI1-e), coeval with established NW Pacific centennial-millennial
productivity/environment cycles, also varied in-phase with Greenland sub-interstadials (Björck
et al., 1998) (Fig. 5). Recent high resolution investigations of Bering Sea sediment cores from
the "Bering Green Belt" (Kuehn et al., 2014) have documented four well-dated laminated
sediment layers during the B/A warming-beginning of the Holocene, with three of them within
the B/A. The synchronicity of the Bering Sea laminated sediment layers with the Greenland sub-
interstadial during B/A warming provides one more piece of evidence supporting the close
atmospheric teleconnection between the N Pacific, EAM, and N Atlantic.
The strongly in-phase linkages between the NW Pacific centennial-millennial
productivity/environment cycles, and the sub-interstadials of summer EAM intensity over GS-
2.1–GI-1 (Figs. 4 and 5) suggest that these abrupt changes in the NW Pacific and EAM have
been forced by similar, or less pronounced, mechanisms to interstadials, such as the shifting of
the ITCZ with the reorganization of atmospheric circulation and the northern westerly jets. In-
phase teleconnection of the NWP/EAM sub-interstadials with those in Greenland was also
observed during LGM-B/A warming. This was weaker during HE 1, which is probably related to
differences in $\delta^{18}$ O between the GISP 2 and NRGIP.

**5.2 The EH**

During the EH the records presented here show a series of abrupt increasing/decreasing
productivity events in the NW Pacific, correlated with sub-interstadials (CsI-EH-1, CsI-EH-2,
CsI-EH-3)/sub-stadials (CsS-EH-1, CsS-EH-2, CsS-EH-3, CsS-EH-4, CsS-EH-5) of the $\delta^{18}$O
records of the Dongge and Hulu caves (Dykoski et al., 2005; Wang et al., 2008) and Greenland
ice cores (North Greenland Ice Core Project members, 2004) (Figs. 4 and 5; Table 2). A visual
comparison with the EAM and Greenland ice core records show synchronicity (positive
correlation) of the increased productivity centennial events in the NW Pacific with the abrupt
warmer climate cycles in Greenland and the summer EAM intensity events, and vice versa over
the EH as well (Figs. 4 and 5). The dated pollen reconstructed the vegetation/climate variability
of south Siberia (Lake Baikal region) (Bezrukova et al., 2011) demonstrated nearly the same
type of centennial-millennial climate variability—confirming their common patterns of change
in the N Hemisphere (N Atlantic, NW Pacific, EAM) over the EH (Fig. 5). Well-dated, high
resolution lithological and geochemical results from the Yanchi playa (NE China) also clearly
show a separation of three sharp cooling events at 8.2 ka, 9.9–10.1 ka, and 11.0–11.2 ka,
synchronous with the cooling shown in the Greenland ice core records (Yu et al., 2006). Yu et al.
(2006) explain this correlation through linkages of the tropical Pacific and the N Atlantic.
Moreover, high resolution geochemical and lithological analyses of the Arolik Lake sediments
(southwestern Alaska) provide evidence that centennial-scale climate shifts during the Holocene
were similar in the sub-polar regions of the N Atlantic and N Pacific (Hu et al., 2003).
These regional climate shifts also occurred concurrently with the periodicities of solar
activity and the production of the cosmogenic nuclides $^{14}$C and $^{10}$Be. The production rates of
these cosmogenic nuclides are negatively correlated with total solar irradiance due to the strength
of magnetic fields embedded into the solar wind. Small variations in solar irradiance could be
responsible for pronounced changes in northern high-latitude climate and environments (Hu et
al., 2003). The variability of sub-polar N Atlantic ice drifting, recorded in the percentage of
hematite-stained grains in the sediment core (Bond et al., 2001), though having lower time
resolution and dating precision compared with production of the cosmogenic nuclides, is
consistent with other centennial climate changes in the N Hemisphere during EH within a timing
precision of 200 years.
Quasi-synchronicity of the changes in the centennial-millennial productivity and
magnetic proxies obtained in the two studied cores, with the sub-interstadials in $\delta^{18}$O records of
Chinese cave speleothem, the Greenland ice cores, and with the nuclide $^{14}$C production during
the EH (Figs. 4, 5), imply that the variability of the NW Pacific climate and environmental
conditions has been strongly related to the EAM and N Atlantic/Greenland climate changes
through atmospheric coupling mechanisms over the studied period of 20–8 ka. In summary, the
NW Pacific results presented here indicate a tight linkage and coherent, persistent pattern of
centennial-millennial scale climate changes in the N Hemisphere over the LGM-EH, which may
serve as a template in high resolution paleoceanography and sediment stratigraphy of the
moderate-high latitudes of the N Pacific.
Since whether N Atlantic-N Pacific climate and hydrological linkages are in-phase or out-
of-phase teleconnections is still debated, empirical data obtained from sediment cores off
Kamchatka allow the provision of an additional test for clarifying this problem at a high
resolution. Previously, it was stated that the N Pacific centennial-millennial productivity/climate
changes are strongly associated with the EAM system variability, which may serve as key
records for the N Pacific due to being the most reliable chronology of the East Asia-N Pacific
region. $\delta^{18}$O records of the GISP2 and NGRIP on the GICC05 age scale (Rasmussen et al., 2014)
may serve as key records for the N Atlantic. The uncertainty in the chronologies of the
Greenland and EAM records is very small (<2%) thus suggesting statistical estimation of their
correlation during the last 25 ka.
Cross correlation (CC) between $\delta^{18}O$ values of Chinese stalagmites (Wang et al., 2008)—
responsible for EAM/N Pacific variability—and NGRIP and GISP2 ice cores (Rasmussen et al.,
2014)—responsible for the Greenland/N Atlantic changes—using moving windows at 1000,
2000, and 3000 years shows their more significant synchronization (from -0.6 to -0.9) during the
period 16.5–8.5 ka (Fig. 6). During earlier (25–16.5 ka) and later (8.5–1 ka) periods there are
differences in CC between the EAM-NGRIP and the EAM-GISP2. However, both CC during
these periods show the occurrence of weak synchronization and/or the absence of significant
correlation (within a range of ±0.25) (Fig. 6). Significant synchronization was also indicated by
CC between EAM-NGRIP during the Middle–Late Holocene. More discrepancies in both CCs
were observed over 19.5–16.5 ka, which may be explained by errors in age measurements and/or
by differences in atmospheric teleconnection between the EAM and the GISP2/NGRIP cores due
to their different locations in Greenland. The statistics imply that the seesaw mechanism between
the EAM/NW Pacific and the Greenland/N Atlantic during 25–1 ka is not effective. However,
they are in line with empirical data of the EAM/N Pacific and the Greenland/N Atlantic
teleconnection by shifting of the westerly jet path (Nagashima et al., 2011; Sun et al., 2012).
**5.3 NW Pacific productivity trends over the LGM-HE 1**
Besides the centennial-millennial productivity/environmental cycles, common NW Pacific
productivity trends are found over the LGM and HE 1 with some differences in other types of
productivity proxies. According to the sharp increase in Antarctic temperature, dust content in
the Greenland ice cores, and significant decrease in the summer EAM, a boundary of LGM/HE 1
was defined at around 17.8 ka (Fig. 5). This is a little earlier than ~17.5 ka, which marks the
beginning of catastrophic iceberg discharges in the HE 1, but nearly coincides with the abrupt
increase of the $^{231}$Pa/$^{230}$Th ratio in the N Atlantic core OCE326-GGC5, which marks the
beginning of the collapse of AMOC (McManus et al., 2004).

During the LGM, most of the productivity proxies demonstrate minimum primary

production in the far NW Pacific without definite trends, although the color b* of core 12KL
shows a small negative trend (Fig. 4). Severe environmental conditions in central Asia, inferred
from vegetation reconstruction (Bezrukova et al., 2011) (Fig. 5), promoted an increase in winter
sea ice covering consistent with high IRD accumulation in the studied region, inferred from CF
and MS records (Fig. 4), that hamper productivity. It is in concord with the established minimum
of productivity in the NW Pacific due to strong stratification preventing the supply of nutrients
required to support productivity in surface waters (Gebhardt et al., 2008).

From 17.8 to 15.3 ka, some productivity proxies of core 41-2—namely TOC and chlorin

associated with the production of calcareous phytoplankton (mostly coccolithophores)—show
significantly increased trends simultaneously to gradual Antarctic warming, accompanied by a
strongly diminishing AMOC (McManus et al., 2004). The diminished AMOC resulted in a major
cooling of the N Atlantic surface water and, most likely, reduced water evaporation in the N
Atlantic and therefore Atlantic-Pacific moisture transport. This condition facilitates a reduction
of precipitation and hence an overall increase of surface water salinity, and decrease of surface
stratification in the N Pacific. This condition promotes an intensification of the intermediate
water ventilation in the N Pacific, and therefore the nutrient supply into the euphotic layer. The
observed trends of productivity proxies are in concord with strong intensification of the
intermediate-depth water ventilation in the N Pacific during HE 1 (Max et al., 2014), based on
the $\delta^{13}$C foraminifera data from the intermediate water and radiocarbon-derived ventilation ages.
However, fairly constant CaCO$_3$ values in both cores (water depth 1924–2145 m) during LGM-
HE 1 do not indicate that the water ventilation penetrated to deep water in the N Pacific over that
time span, because carbonate concentration in the sediment is strongly defined by the ventilation
of bathed water (Yu et al., 2013). While the productivity proxies Si-bio and color b*, associated

with siliceous phytoplankton production (mostly diatoms), do not show significant trends during

HE 1 up to ~15.3 ka, the strong sea ice effect with high IRD input up to 15.3 ka, shown by CF

and MS records, (Figs. 2 and 4) was significant in the studied area and probably overwhelmed

the production of diatom algae for coccolithophores, due to a large spring–early summer surface

water stratification during seasonal sea ice melting.

A sharp increase in NW Pacific primary production, and a rise in diatom production since

~15.3 ka, indicated by most productivity proxies and Si-bio and color b* records with a

culmination at sub-interstadial GI1-e of B/A warming, was likely induced by a decrease in sea

ice influence and its spring melting, favoring a weakening of surface stratification (Figs. 4 and

2). The timing of the decrease in the sea ice cover since ~15.3 ka is consistent with the surface

water warming (Max et al., 2012), and with the central Asian vegetation/environment

amelioration inferred by Bezrukova et al. (2011) from pollen reconstructions (Fig. 5). Such a

pattern of productivity changes in the N Pacific and the Bering Sea during glacial/interglacial

transitions has been observed in other cores (Caissie et al., 2010; Galbraith et al., 2007; Gebhardt

et al., 2008; Keigwin, 1998) and was likely a persistent feature for the N Pacific and its realm,

forced by the resumption of the AMOC at the B/A warming coeval with the cooling in

Antarctica (Fig. 5). In the Okhotsk Sea, the beginning of the diatom production and

accumulation of the diatomaceous sediments had begun only in the Middle Holocene (5–6 kyr

BP) due to the later reduction of sea-ice cover, and later breakdown of spring/early summer

surface water stratification (Gorbarenko et al., 2014).

**6. Conclusion**

This study presents high resolution records of a suite of productivity proxies (TOC,

$CaCO_3$, chlorin, color b*, Ba-bio, Br-bio, Si-bio), sediment lithological (CF), and magnetic

properties (PM, MS, and RPI) from sediment core 41-2, taken from the NW Pacific (East

Kamchatka slope). Results presented here reveal a sequence of 13 centennial-millennial scale

regional productivity increase/environment amelioration events over the LGM-EH (20–8 ka) in
the far NW Pacific.
The age model of core 41-2 was constructed by using available AMS [14]C dating, with
more age control points identified by correlating the centennial-millennial productivity events,
RPI, and PM of the core with those of the well-dated nearby core 12KL (Max et al., 2012, 2014).
Thus, all available AMS [14]C dating of core 12KL was projected successfully to core 41-2. Based
on putting all radiocarbon data of both cores on the $\delta^{18}$O record of the Chinese cave stalagmites
(Wang et al., 2008), the close time correlation of NW Pacific productivity events with sub-
interstadials in the summer EAM over the period 20–8 ka was inferred and used for further fine
age model construction. Three NW Pacific abrupt productivity increase events are strongly
linked to CsIs during the LGM (20–17.8 ka); three during HE 1 (17.8–14.7 ka), four during B/A
warming, and three over the EH.
The reconstruction in this paper suggests that the NW Pacific centennial-millennial
productivity increase and the summer EAM intensification events are positively correlated with
Greenland abrupt warmings, indicating a strong atmospheric teleconnection between the N
Pacific and the N Atlantic, most likely due to the ITCZ shifting and the reorganization of the
northern westerlies. This echoes the mechanism proposed in previous studies for the N
hemisphere interstadials and stadials (Caissie et al., 2010; Kienast and McKay, 2001; Max et al.,
2012; Riethdorf et al., 2013). Especially highlighted here is the fact that a comparison of the NW
Pacific centennial-millennial productivity events/EAM sub-interstadial with $\delta^{18}$O records of the
EDML ice core over glaciation and deglaciation suggests a Southern Hemisphere "push" effect
on the boreal summer EAM propagation.
During the LGM the results indicate productivity minima that are consistent with
previous observations in the NW Pacific and severe vegetation/climate conditions in central Asia
(Bezrukova et al., 2011). Therefore, strong regional sea ice covering is consistent with the
hypothesis that a strong stratification prevented the supply of nutrients required for supporting
productivity in surface waters (Gebhardt et al., 2008). The productivity proxies associated with
calcareous phytoplankton productions show increased trends from 17.8 to 15.3 ka. These trends
share the same structure of change with the gradual Antarctic warming accompanied by a
significantly diminished AMOC (McManus et al., 2004). The cooling of the N Atlantic surface
water reduced water evaporation in the N Atlantic, as well as Atlantic-Pacific moisture transport.
This, in turn, facilitates the increased surface water salinity and decreases surface stratification in
the N Pacific. The weakening stratification further intensifies the intermediate water ventilation
in the N Pacific and the supply of nutrients into the euphotic layer. It is especially noted that a
sharp increase of NW Pacific primary production since around 15.3 ka was indicated by nearly
all productivity proxies, accompanied by some climate warming and a decrease in sea ice cover.
Subsequently, a strong productivity spike of sub-interstadial GI-1e at beginning of the B/A
warming is associated with a resumption of the AMOC and the further decrease of sea ice
influence, accompanied by a rise in diatom production.

The synchronicity in changes of the NW Pacific centennial-millennial productivity events

with the sub-interstadials in $\delta^{18}O$ of Chinese stalagmites calcite, Greenland ice cores, and with
the nuclide $^{14}C$ production during the EH (Figs. 4 and 5) imply that the variability of the NW
Pacific climate is strongly linked to the summer EAM and N Atlantic/Greenland climate
changes. The linkage is likely driven effectively by atmospheric coupling mechanisms forced by
variations in solar irradiance. Regardless of what specific driving mechanisms are responsible for
the teleconnection, strong causal linkages of the centennial-millennial productivity/climate
variability in the NW Pacific with sub-interstadials of summer EAM from the LGM to EH
reported here is a persistent feature of high resolution, far NW Pacific paleoceanography and
sediment stratigraphy, and is almost synchronous with the Greenland/N Atlantic short-term
changes.
**Acknowledgements**
We are grateful to Drs. Ralf Tiedemann and Dirk Nürnberg (AWI, GEOMAR, Germany) for
a long and fruitful cooperation, and for providing samples and the dataset of core 12KL. We are
indebted to Dr. John Southon (USA) for the AMS $^{14}$C dating.
This research work was supported by the RFBR (Russia Fund of Basic Research), Russia project
(13-05-00296a, 16-55-53048, and 16-05-00127), Russian Federation budget No 01201363042,
National Natural Science Foundation of China projects (41420104005, 40710069004, and
40431002), and the Russia-Taiwan Research Cooperation projects 14-HHC-002 and 17-MHT-

003.

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

TABLES

Table 1. AMS $^{14}$C data on monospecies planktonic foraminifera *N. pachyderma* sin. and

benthic foraminifera *Epistominella pacifica* and *Uvigerina parvocastata* of core 41-2. All
measured $^{14}$C age data were corrected by NW Pacific surface water reservoir ages of 900 years
(Max et al., 2012). In case of using benthic foraminifera we accept difference in coeval benthic-
planktic foraminifera ages equals to 1400 years for depth water 1940 m, based on the
unpublished datum and results of Max et al. (2014). All radiocarbon ages were converted into
calibrated 1-sigma calendar age using the calibration program CALIB REV 7.0.1 (Stuiver and
Reimer, 1993) with the Marine13 calibration curve (Reimer et al., 2013).

| # | Lab. code | core depth cm | foraminifera species | $^{14}$C-age year | Err.1 sigma year | calendar age, ka |
|---|-----------|---------------|----------------------|-------------------|------------------|------------------|
| 1 | YAUT-021713 | 120 | *E. pacifica* | 10078 | 47 | 9.121 |
| 2 | YAUT-021714 | 127.5 | *E. pacifica* | 10340 | 42 | 9.445 |
| 3 | UCIAMS-148095 | 298 | *N. pachyd.* | 13160 | 50 | 14.393 |
| 4 | UCIAMS-148096 | 156 | *Uv. parvoc.* | 11135 | 45 | 10.60 |
| 5 | UCIAMS-148098 | 306 | *Uv. parvoc.* | 14185 | 35 | 14.616 |


Table 2. Centennial-millennial productivity increase/environment amelioration events
over 25-8 ka ago plus abrupt productivity drop/cooling Events during Early Holocene in the NW
Pacific core 41-2 which had occurred in-phase with Chinese sub-interstadials (CsI) of the
summer EAM intensification and Chinese sub-stadials (CsS) of winter EAM activation.

| Events | Core interval, cm | Averaged cal. age, ka |
|--------|-------------------|-----------------------|
| CsS-EH-1 | 106-110 | 8.2 |
| CsS-EH-2 | 117-123 | 9.2 |
| CsI-EH-1 | 125-132 | 9.8 |
| CsS-EH-3 | 138-143 | 10.2 |
| CsI-EH-2 | 148-153 | 10.7 |

| | | |
|---|---|---|
| CsS-EH-4 | 155-159 | 10.95 |
| CsS-EH-4' | 162-167 | 11.15 |
| CsI-EH-3 | 168-181 | 11.4 |
| CsI-GI1-a | 233-243 | 13.05 |
| CsI-GI1-c1 | 248-262 | 13.5 |
| CsI-GI1-c3 | 268-278 | 13.8 |
| CsI-GI1-e | 291-312 | 14.45 |
| CsI-GS2.1-1 | 335-340 | 15.45 |
| CsI-GS2.1-2 | 355-362 | 16.55 |
| CsI-GS2.1-3 | 375-383 | 17.56 |
| CsI-GS2.1-4 | 388-395 | 18.1 |
| CsI-GS2.1-5 | 400-410 | 18.85 |
| CsI-GS2.1-6 | 431-447 | 19.8 |


Table 3.The key time points of core 41-2 based on the available AMS [14]C data of core
41-2, projection of AMS [14]C data of core 12KL on the core 41-2 depth according to correlation
of related increased productivity events and RPI records plus correlation of the productivity
events with related sub-interstadials of the highly resolved, absolutely dated E Asia monsoon
(Wang et al., 2008) beyond the projected [14]C data. AMS [14]C datum of core 12KL and age at
depth of 706 cm was accepted according to the Tiedemann/Max age model 2 (Max et al., 2012,

2014).


| Depth | AMS [14]C core 41-2 | Key time points of core 12KL | correlation with ages of China sub-interstadial | Accepted key time points |
|---|---|---|---|---|
| cm | cal. age, ka | age, ka/ depth (cm) | age, ka/CsI | cal. age, ka |
| 120 | 9.12 | | | 9.12 |
| 127.5 | 9.45 | | | |
| 126 | | 9.51/210 | | 9.51 |
| 156 | 10.6 | | | |

| | | | | |
|---|---|---|---|---|
| 159 | | 11.08/295 | | 11.08 |
| 167 | | 11.31/340 | | 11.31 |
| 239 | | | 13.0/CsI-GI1-a | 13.0 |
| 251 | | 13.42/508 | | 13.42 |
| 273 | | 13.79/550 | | 13.79 |
| 298 | 14.39 | | | |
| 303 | | 14.42/611 | | 14.42 |
| 306 | 14.61 | | | |
| 337 | | | 15.42/CsI-GS2.1-1 | 15.42 |
| 348 | | 16.16/706 | | 16.16 |
| 357 | | | 16.51/CsI-GS2.1-2 | 16.51 |
| 379 | | | 17.56/CsI-GS2.1-3 | 17.56 |
| 393 | | | 18.12/CsI-GS2.1-4 | 18.12 |
| 402 | | 18.6/821 | | 18.6 |
| 431 | | 19.54/876 | | |



FIGURE CAPTIONS

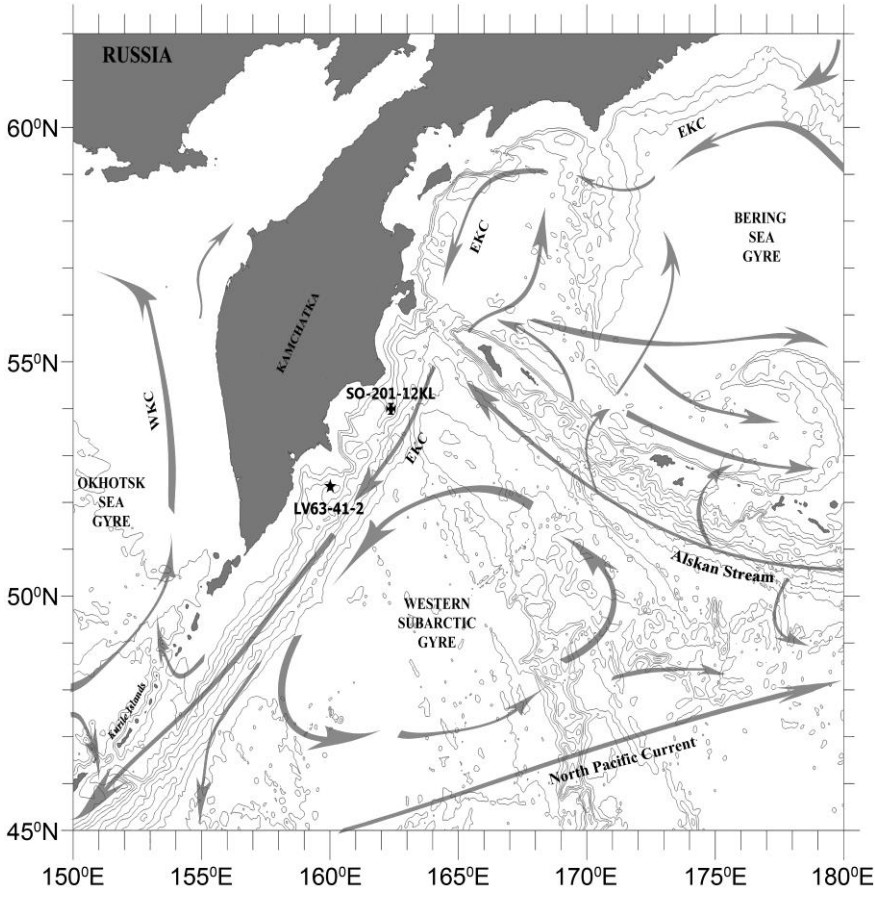


Fig. 1. Bathymetry, surface water currents, and location of cores 41-2 (star) and 12KL (cross)
(Max et al., 2012) in the North Pacific. Surface currents as in Favorite et al. (1976) with
modifications. EKC—East Kamchatka Current, WKC—West Kamchatka Current.

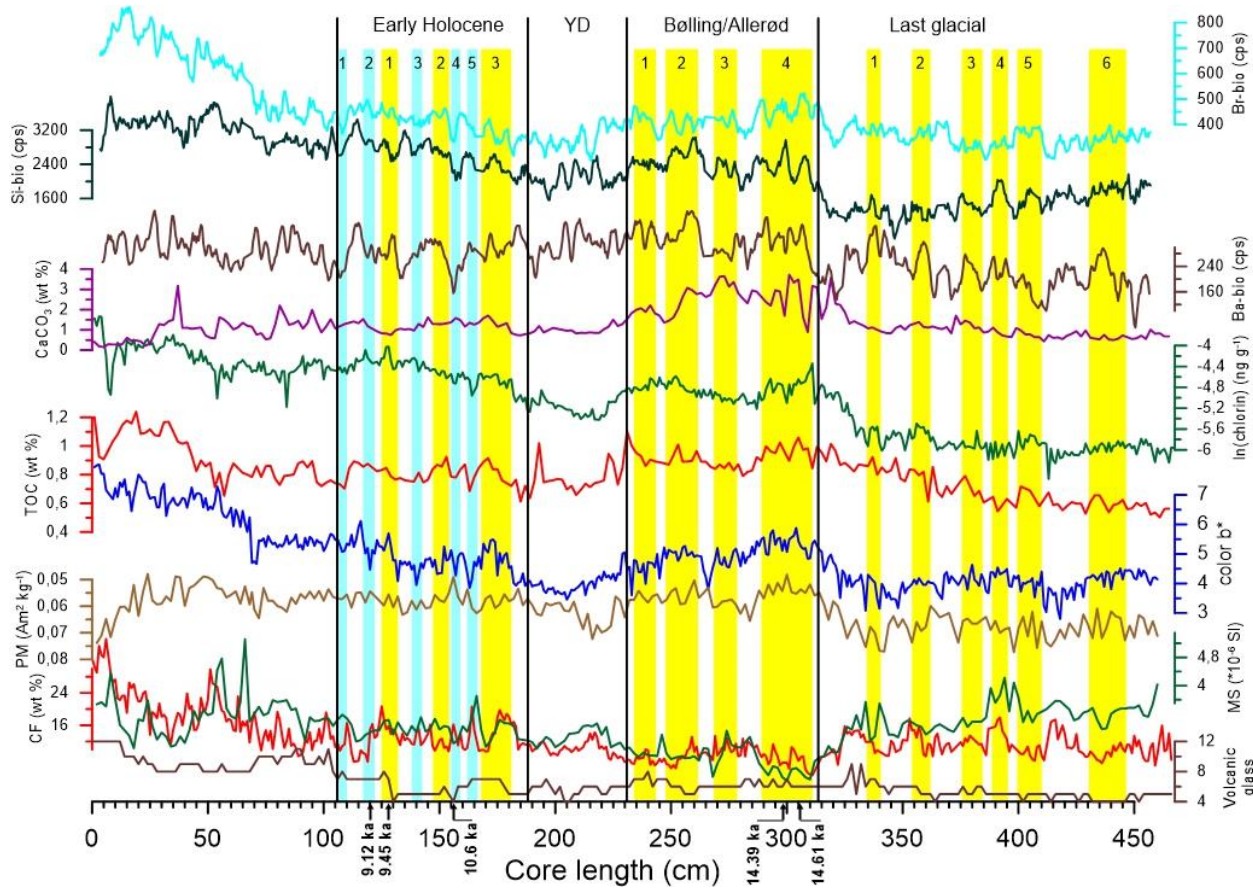


Fig. 2. Records (from bottom to top) of the share of volcanic grains in the sediment fraction >
150 μm, weight percentages of the CF, magnetic susceptibility (MS), paramagnetic
magnetization (PM), color b*; TOC, chlorin, $CaCO_3$, Ba-bio, Si-bio (opal), and Br-bio content
versus core 41-2 depth. Preliminiary boundaries of B/A warming, YD cooling, and Holocene are
shown according to total reguliarities of productivity variability in the NW Pacific, Sea of
Okhotsk, and Bering Sea (Galbraith et al., 2007; Gorbarenko, 1996; Gorbarenko and Goldberg,
2005; Keigwin, 1998; Seki et al., 2004); and AMS [14]C data (calendar ka) shown at the
base.Yellow (blue) bars depict the centennial-millenial increased productivity/environmental
ameloiration (cooling) events according to most productivity proxies and decreases in PM.

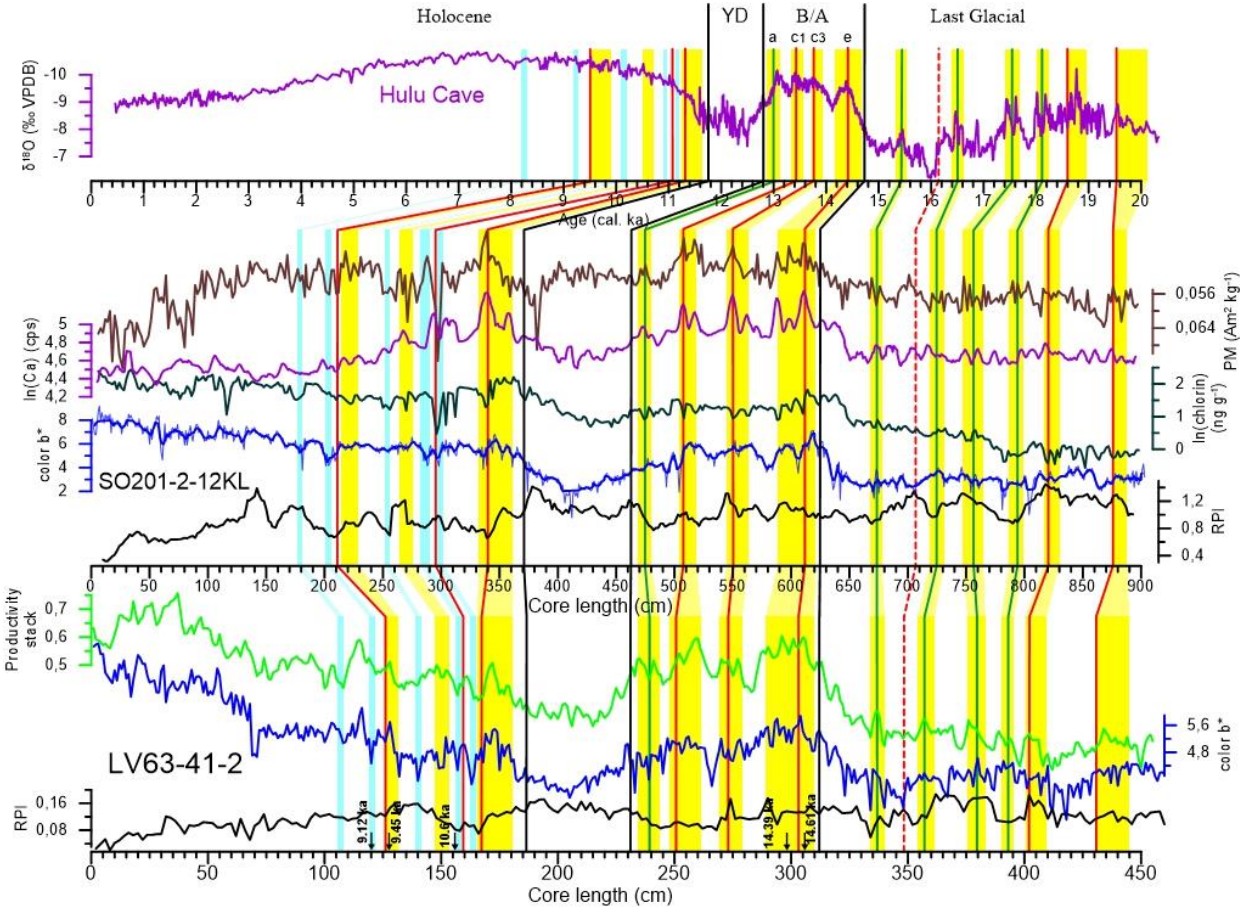


Fig. 3. Correlation of the increased productivity event cycles in core 41-2 (lower panel) with those in 12KL (middle panel) versus depth with sub-interstadials of the $\delta^{18}O$ calcite of Chinese stalagmites (Wang et al., 2008) (upper panel). Productivity cycles for cores 41-2 are based on the stack of productivity proxies and PM records (Fig. 2) and 12KL (Ca, chlorin, color b*, and PM records) were correlated according to sychronous changes in productivity proxies, paramagnetic magnetization, magnetic relative paleomagnetic intensity (RPI), and $^{14}C$ AMS data of both cores. AMS $^{14}C$ data of core 41-2 is shown at the base. According to the correlation of the productivity cycles and curves of RPI, the red lines are related to key time points of core 12KL (middle panel) and the green lines with the relative Chinese sub-interstadials of the $\delta^{18}O$ calcite of Chinese stalagmites (Wang et al., 2008) (upper panel) were projected into corresponded depths of core 41-2 (bottom panel). Yellow (blue) bars depict the centennial-millenial increased productivity/environmental ameloiration (cooling) events according to most productivity proxies and decreases in PM.

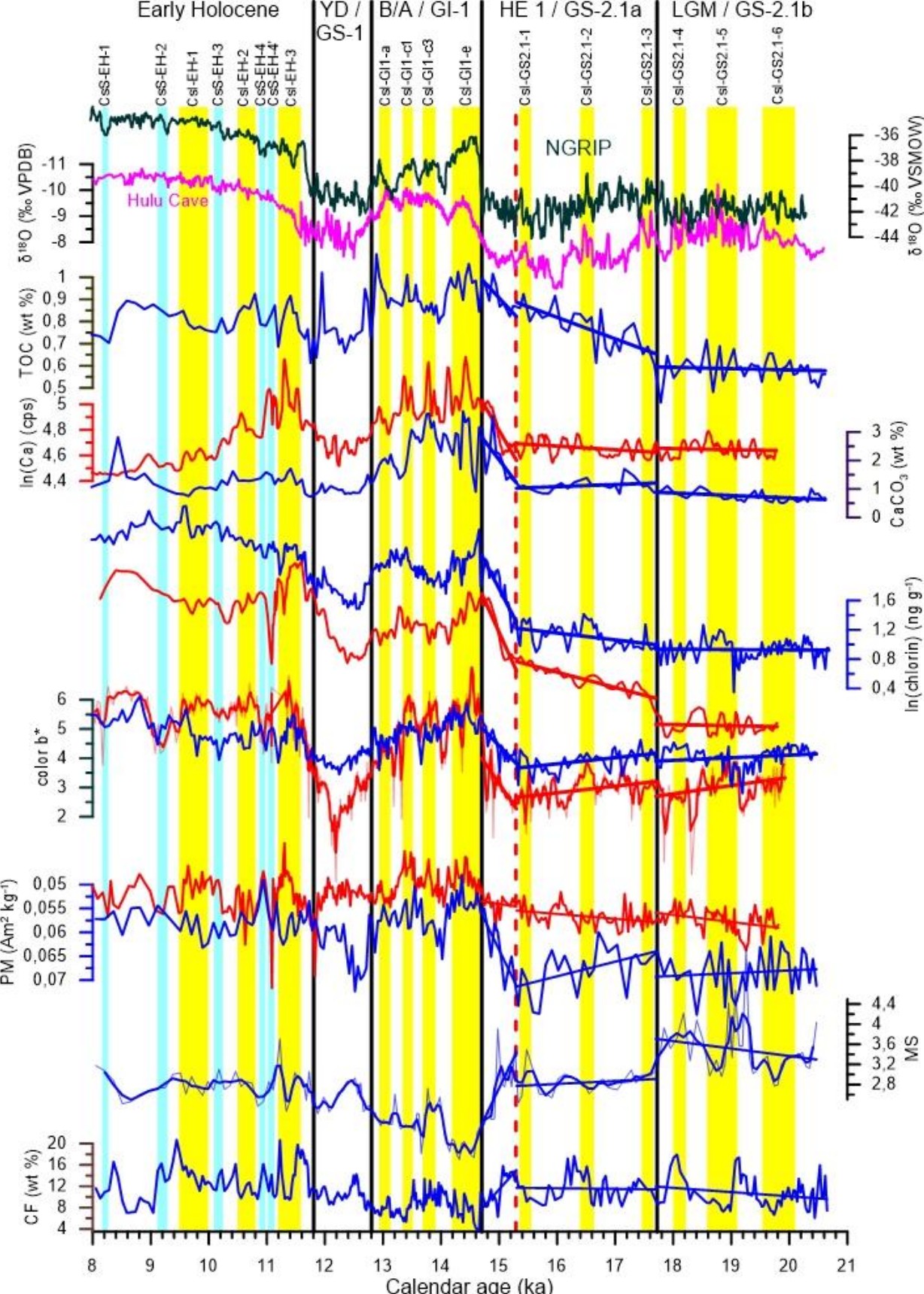

Fig. 4. High resolution variability of the productivity and lithologic proxies in the NW Pacific
(off Kamchatka) over the 21–8 ka period. CF percentages, MS, paramagnetic magnetization and
color b*, chlorin, $CaCO_3$, TOC content determined in cores 41-2 (blue lines) and 12KL (red
lines) are shown from bottom to top. The NW Pacific centennial-millennial productivity cycles
characterized by an increase in most productivity proxies are clearly associated with the abrupt
summer EAM intensification revealed in the Chinese cave stalagmites, defined as sub-
interstadial, and less are pronounced with short-term events in the Greenland ice core $\delta^{18}O$
records. Linear trends are shown for the productivity indices over LGM and HE 1. Yellow (blue)
bars depict the centennial-millenial increased productivity/environmental ameloiration (cooling)
events according to most productivity proxies and decreases in PM.

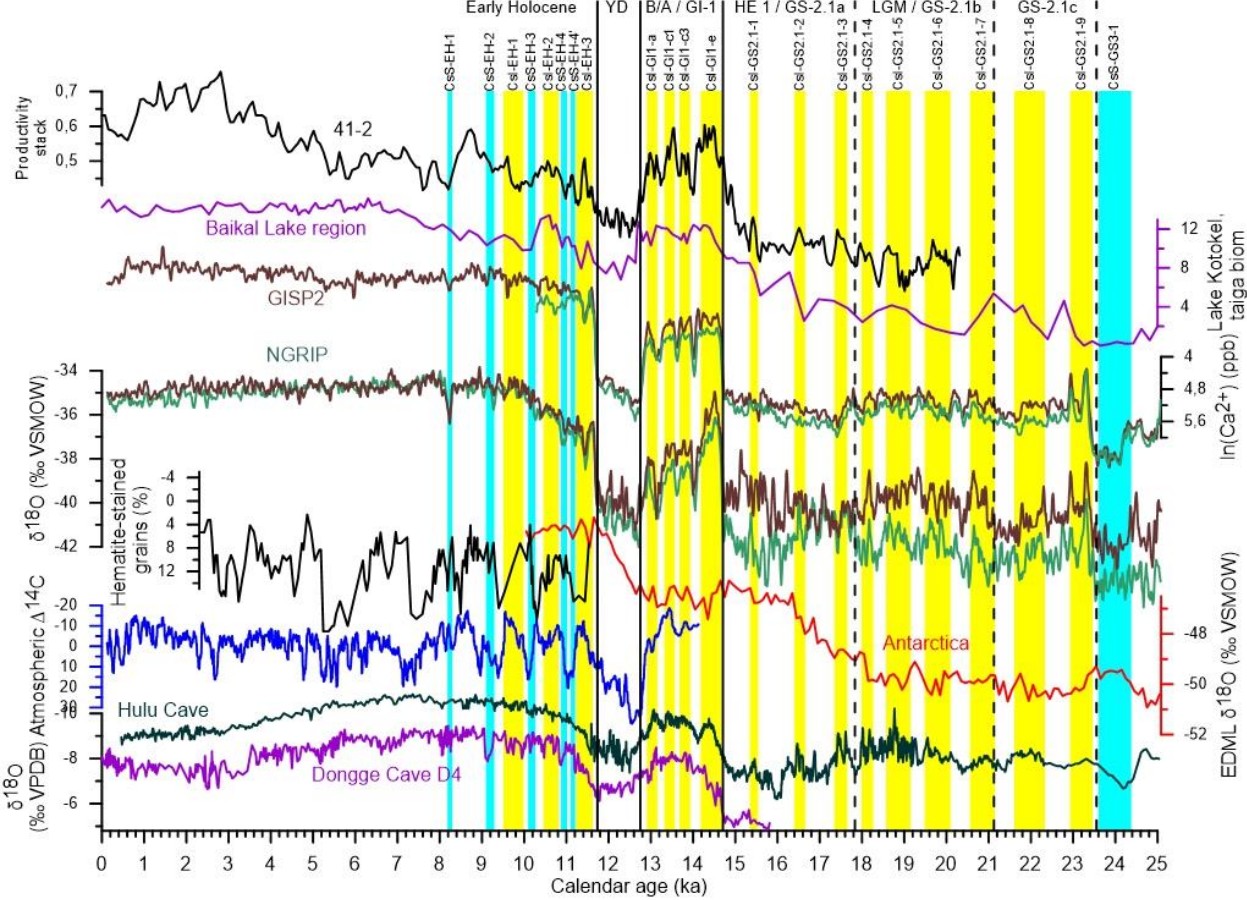

Fig. 5. Compilation of N-S hemisphere milestone climate records, solar activity, NW Pacific
productivity cycles, and Southern Siberian environment during the last 25 ka. From bottom to
top: absolutely dated $\delta^{18}O$ calcite of Chinese cave stalagmites (Dykoski et al., 2005; Wang et al.,
2008) characterized EAM activity; the residual atmospheric $\Delta^{14}C$ record of around 2000-year
moving average (Reimer et al., 2004) indicated solar irradiance variability; oxygen isotope
EDML records after methane synchronization with the North Greenland ice core (EPICA
Community Members, 2006); the petrologic tracer of drift ice in the N Atlantic (Bond et al.,
2001); the $\delta^{18}O$ and $Ca^{2+}$ records in the Greenland NGRIP and GISP 2 ice core indicated air
temperature and dust variability on GICC05 age scale (Rasmussen et al., 2014), pollen
reconstructed Southern Siberia environment changes (Lake Kotokel, Lake Baikal region)
(Bezrukova et al., 2011); and productivity stack for core 41-2. Yellow (blue) bars depict the
centennial-millenial increased productivity/environmental ameloiration (cooling) events. NW
Pacific centennial-millennial productivity cycles are accompanied by interstadial and sub-
interstadial intensification of the summer EAM over 25–8 ka, and increase of solar irradiance
during B/A and EH short term warmings. Their correlation with short term increased Greenland
temperature (NGRIP ice core) and a decreased Antarctic temperature are less pronounced but
seem to be marked as well.

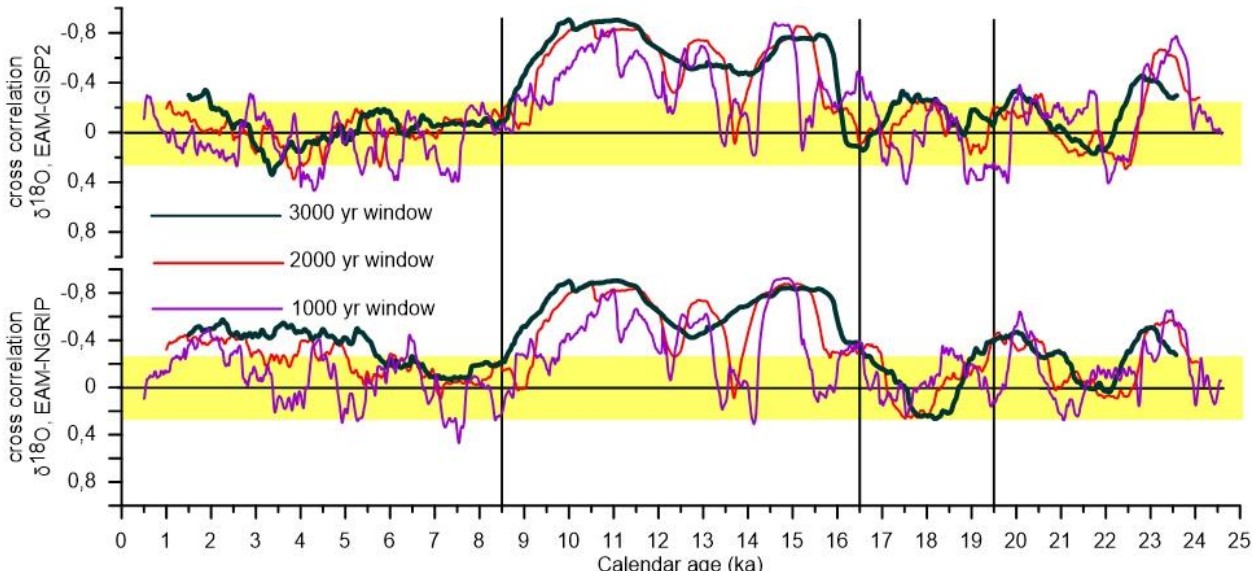

Fig. 6. Cross correlation of the EAM and Greenland climate variability calculated by correlation
of $\delta^{18}O$ values of the calcite of Chinese stalagmites (Wang et al., 2008) with those of the NGRIP
(lower panel) and GISP 2 (upper panel) ice cores (Rasmussen et al., 2014) using moving
windows of 1000 years (purple lines), 2000 years (red lines), and 3000 years (green lines) over
the last 25 ka. Yellow bars show areas with insignificant cross correlation ranging between +0.25
and -0.25. Cross correlation between the EAM and Greenland using a moving window of 3000
years is negative during the period 16.5–8.5 ka, and insignificant or weakly negative during
earlier and later periods of 25–16.5 ka and of 8.5–0 ka confirmed the EAM and the Greenland
synchronicity.