# Peer review of "Centennial to millennial climate variability in the far northwestern Pacific (off"

_Climate of the Past, 2016_

## Referee Comment (RC1) · Anonymous Referee #1 · 1 Dec 2016

This study presents high-resolution proxy data derived from a marine sediment record of the Northwest Pacific. Based on lithophysical- and productivity proxies (e.g. magnetic properties, TOC content or chlorine measurements) the authors provide time-series of millennial- to centennial-scale North Pacific climate variability from the Last Glacial Maximum (LGM) into the Early Holocene (EH). Several short-term (centennial-scale) productivity changes/cycles have been identified from 20 to 8 ka and are compared to high-resolution records reflecting changes in the East Asian Monsoon (EAM)

system as well as climate variability recorded Greenland- and Antarctic ice cores. The authors suggest a close relationship between changes in productivity in the northwestern Pacific, changes in EAM and Greenland/North Atlantic climate variability and further speculate about underlying mechanisms to explain North Atlantic - North Pacific climate synchronicity on centennial timescales.

General comments:

As there is currently quite some debate about the relationship of North Atlantic and North Pacific climate variability during the last glacial termination, the study of Gorbarenko et al. is a timely and relevant research topic. Unfortunately, problems regarding the English (grammar and usage) affect the clarity of the writing and sometimes it is quite difficult to follow authors' interpretations. Moreover, there are several issues that need to be discussed before the manuscript should be considered for publication in Climate of the Past. Please find below my recommendations to improve the quality of the manuscript:

I have some major complaints about the construction of the age model for core 41-2. The first serious concern is about the use of AMS 14C ages derived from benthic foraminifera (E.pacific and U.parvoc.) to establish age control points for core 41-2. More specifically, a constant offset of 1400 years between planktic and benthic foraminifera has been used to correct benthic ages for reservoir effects according to unpublished? and total regional results? of Max et al. (2014). However, looking into the original publication of Max et al. (2014) reveals that planktic – benthic age differences (BP ages) in core 12KL (used for correlation) varied substantially during the last glacial termination and the Holocene and are not constant through time. The variability in BP ages given in core 12KL is thus in conflict with age assumptions made on benthic foraminifers in this study. Moreover, there is some confusion about correlation of core 41-2 to neighbor core 12KL. Several AMS 14C ages have been transferred from core 12KL to 41-2 to improve the stratigraphy. However, it has been stated by the authors that one age dating from core 12KL was not transferred to 41-2 (16.53 ka at 695cm

in core 12KL) but in the original age model of core 12KL the age in 695cm is 15.9 ka and not 16.53 ka! I feel that the current age model needs to be revised and further improved. To improve the age model, it would make sense to e.g. check for correlation to well-dated core SO202-7-6 (Serno et al., 2015). This core has an excellent age model consisting of > 40 AMS 14C ages that, in principle, could be transferred to core 41-2.

A major point that has been made is that the NW-Pacific experienced a sequence of millennial – centennial-scale productivity changes during the last 21ka. The authors further suggest an in-phase behavior between changes East Asian Summer Monsoon and to some extent links to Greenland sub-interstadials. I suggest adding another table, which should give the reader information about the occurrence and length of proposed events. I also recognized that cross-correlation has been done to compare time-series of the EAM and Greenland ice core data. What I'm missing here is cross correlation between 41-2 and Greenland and EAM records. I strongly recommend cross-correlation between 41-2 and Greenland and EAM records because this could, from a statistical point of view, strengthen the main conclusion of this study. Ideally, the time window for cross-correlation should be reduced (< 1000 years) according to the centennial-scale variability in 41-2 and other climate records.

Specific comments/Technical corrections:

Abstract:

Line 17: "The core age model" sounds strange. Better is "The age model of core 41-2 is. . .

Line 19: Please replace "SO-201-2-12KL" by "SO201-2-12KL"

Introduction:

Line 38-41: Some references (e.g. Kienast et al., 2001 or Seki et al., 2002) are related to studies in the Northeast Pacific. Accordingly, "NW Pacific" should be replaced by "North Pacific"

Line 42: A more proper citation would be Kühn et al. (2014)

Line 49: "atmospheric teleconnections" Please elaborate a little bit more in the Introduction the mechanisms behind atmospheric coupling.

Line 52: ...Riethdorf et al. (2013) further suggest...

Line 57-61: This sentence is quite long and should be subdivided into two sentences.

Line 77: "With our methodologically robust age controls"...please see major complaint about the age model above. It is probably more accurate to say that the age model (in principle) allows to investigate centennial-millennial scale productivity...

Material and methods:

Line 87: Is there any citation (cruise report?) available???

Line 106: Please replace "computing" by "calculating"

Line 109-110: A better beginning of these sentence would be: "It has been shown that variability in color b*...."

Line 112-124: Please see comments above regarding the robustness of the age model. Do you use the Intcal09 and Marine09 dataset as stated in Line 123-124 or Marine13 calibration curve as stated in Table 1???

Results:

Line 180-183: This sentence is confusing. Please clarify.

Line 200-225: This chapter ("Age model") describes the correlation to neighbor core 12KL (please see major comments above) and EAM records. I think the whole chapter should be shifted to the beginning of the Results followed by the paragraph describing other parameter (productivity proxies, CF etc.) derived from sediments. At this point it would make sense to give ages (onset and duration) of the centennial-scale productivity events (e.g. by adding another Table; see main comments above)

Discussion:

Line 243-285: This chapter deals with centennial-scale NW-Pacific productivity events and linkages to EAM records and Greenland ice core data. Thus, I suggest to modify the name of this chapter from "N-S hemispheres climatic linkages of...." to "Northern Hemisphere climatic linkages...". However, studied records are also compared to EPICA data (from the beginning of line 286) and it would make sense to shift this information to another chapter named e.g. "Southern Hemisphere influence on centennial-millennial scale productivity....". There is also other proxy-data available from core 12KL (e.g. SST data; Max et al., 2012 or sea-ice variability data; Méheust et al., 2016), which should be discussed. Accordingly, it could make sense to add SST and/or sea-ice variability data from core 12KL to Figure 5, which serve as additional evidence for millennial-scale variability in the NW Pacific.

Line 259: missing citation (e.g. Ruth et al., 2007)

Line 278-279: "stronger winter monsoon supplies more dust to the loess plateau". Please clarify.

Line 355-410: These chapters deal with NW Pacific productivity trends over the LGM-HE1 and the Early Holocene. I suggest moving these chapters to the beginning of the Discussion, followed than by discussion of possible mechanisms to explain productivity events (e.g. one chapters describing links to Northern Hemisphere/Southern Hemisphere).

Line 379-384: How does enhanced intermediate water ventilation promotes more nutrients to the euphotic layer? Most studies in the NW Pacific found substantially reduced nutrients/productivity during HS1 (e.g. Riethdorf et al., 2013). The enhanced formation of intermediate water during HS1 has been related to strong expansion of sea-ice and brine rejection as main processes to intensify mid-depth ventilation (e.g. Rella et al., 2012).

Line 413-415: Millennial-scale climate events are also well known from the North Atlantic during the Holocene (Bond cycles; Bond et al., 1997). Is there any relationship to millennial-scale events in the North Atlantic?

Figure 1: The core locations are hardly visible in this figure. Please modify.

Figure 3:Please remove all datasets, which are not necessary for age model construction. Please blow up a little bit more the y-axis to help the reader identifying variability in the given records, which is hard to recognize in the current figure.

Figure 4: Please see comments to Figure 3.

Figure 5: It would make sense to implement one proxy dataset from core 41-2 (e.g. Si-bio) for comparison.

References:

Bond, G., W. Showers, M. Cheseby, R. Lotti, P. Almasi, P. deMenocal, P. Priore, H. Cullen, I. Hajdas, and G. Bonani (1997), A pervasive millennial-scale cycle in North Atlantic Holocene and glacial climates, Science, 278(5341), 1257-1266.

Max, L., L. Lembke-Jene, J. R. Riethdorf, R. Tiedemann, D. Nurnberg, H. Kuhn, and A. Mackensen (2014), Pulses of enhanced North Pacific Intermediate Water ventilation from the Okhotsk Sea and Bering Sea during the last deglaciation, Climate of the Past, 10(2), 591-605.

Meheust, M., R. Stein, K. Fahl, L. Max, and J. R. Riethdorf (2016), High-resolution IP25-based reconstruction of sea-ice variability in the western North Pacific and Bering Sea during the past 18,000 years, Geo-Mar Lett, 36(2), 101-111.

Rella, S. F., R. Tada, K. Nagashima, M. Ikehara, T. Itaki, K. Ohkushi, T. Sakamoto, N. Harada, and M. Uchida (2012), Abrupt changes of intermediate water properties on the northeastern slope of the Bering Sea during the last glacial and deglacial period, Paleoceanography, 27.

Interactive
comment

[Figure]

Ruth, U., M. Bigler, R. Rothlisberger, M. L. Siggaard-Andersen, S. Kipfstuhl, K. Goto-Azuma, M. E. Hansson, S. J. Johnsen, H. Y. Lu, and J. P. Steffensen (2007), Ice core evidence for a very tight link between North Atlantic and east Asian glacial climate, Geophysical Research Letters, 34(3).

Serno, S., G. Winckler, R. F. Anderson, E. Maier, H. J. Ren, R. Gersonde, and G. H. Haug (2015), Comparing dust flux records from the Subarctic North Pacific and Greenland: Implications for atmospheric transport to Greenland and for the application of dust as a chronostratigraphic tool, Paleoceanography, 30(6), 583-600.

---

## Referee Comment (RC2) · Anonymous Referee #2 · 4 Jan 2017

General Comments

This paper presents a very high-resolution record of productivity in the Northwest Pacific for the past 21 kyrs. It is really a remarkable amount of data especially for the discretely sampled analyses (TOC, grain size, etc). The paper attempts to answer the question of whether productivity changes in the North Pacific were synchronous or asynchronous from changes in North Atlantic climate. They conclude that increased productivity in the North Pacific is positively correlated with warm intervals in Greenland

and attribute this to "tight atmospheric teleconnections."

There are several major issues with this manuscript that make it difficult to follow the authors' conclusions. Specifically, issues with English grammar and word choice make it very difficult to understand at times, the age model causes extreme concern, and the assignment of high productivity/warm events appears to be arbitrary for the core studied (41-2).

I highlight a few issues with the word choice and organization of the paper below in "Technical Corrections," however this should not be considered an exhaustive list as I tried to avoid copy-editing as much as possible.

It is possible that I have misunderstood how the authors constructed the age model, however it appears that they first radiocarbon dated a mix of benthic and planktonic foraminifera. They then applied a correction of 1400 years to the benthic foraminifera based on the data in Max et al., (2014). They then compared productivity cycles in 41-2 and 12KL to refine their age model and discarded four of the five radiocarbon ages from core 41-2. Finally, they correlated their productivity cycles to well-dated Chinese subinterstadials.

This presents serious concerns for several reasons:

1. It's unclear why the authors are using benthic foraminifera at all. Are planktonics not plentiful enough? Why weren't radiocarbon samples chosen from the $CaCO_3$ peaks? Surely there should be enough planktic foraminifera when carbonate is high.

2. It's unclear at which point the authors apply the ventilation conversion– before or after calibrating–after would be appropriate.

3. It's unclear why they chose 1400 as a stable ventilation age. One of the main points of Max et al., 2014 is that ventilation changes dramatically and frequently during deglaciation. In that paper it ranges from 160 to more than 2500 years. It might have been more appropriate to use ventilation estimates corresponding to the approximate

calibrated age.

4. It's unclear how the productivity cycles are determined in 41-2. The authors state that they're "based on a suite of productivity proxies and PM records" and that they correspond to synchronous changes in these proxies. However, I have examined several of these intervals and cannot find the commonalities between them. As I understand it, the authors are using Ba, Br, Si, b*, TOC, CaCO3, and chlorin as productivity proxies. Looking first at event 6 from the Last Glacial, I see that some of these proxies are flat during this event, some are fluctuating, and some are high. This pattern continues for all the other productivity events as well. Barium seems to most often be high during productivity events, but there are several peaks in Barium that are not associated with productivity events—why not? Clarifying the determination of these events is essential to all aspects of this manuscript.

5. It's unclear how the authors determined which radiocarbon ages to discard. They discarded three benthic ages and the only planktic age measured. The planktic age is probably the strongest part of the age model and it seems to fit in with their correlation to 12KL. Why is it discarded? Why is one benthic age kept, but not others?

6. The final correlation of the productivity events to Chinese subinterstadials is perhaps the most troubling part of this age model. If the age model for 41-2 is tuned to the oxygen isotopes from Chinese caves, then the authors can not claim that productivity events in the North Pacific happened synchronously with these sub-interstadials. This is circular logic.

In addition, it would be useful to include a discussion that addresses the differences between the myriad productivity proxies. By no means do these records look the same, especially on centennial to millennial scales. Why not?

Specific Comments

Section 2.1: It's unclear how tephra was estimated. Was it identified under a microscope? By magnetic susceptibility, some kind of Principal Component Analysis of the XRF scanning? I'm not sure what "semi-quantitative component analysis of this fraction with a total of 12 ranged scales" means.

When I first read Section 2.4, I was under the impression that this manuscript would present planktic-benthic pairs of radiocarbon dated foraminifera. I needed to examine the table myself to determine that it does not. This should be clarified.

In Section 2.6, it's unclear how the terrigenous component of Ba, Br, and Si was determined. Lines 159-160 read, "The terrigenous component, in turn, was calculated from empirical regional (Ba/Al)ter ratios in the sediment core with the lowest Ba-tot contents." What sediment core are you referring to? Where is it located? What is the regional observed value (Ba/Al)ter? If this is an empirical value, it should not be vague.

The age model should come before any discussion of sediment ages in the Results Section, i.e. Sections 3 and 4 should be reversed (and revised accordingly).

In line 169, the authors posit that they observe high productivity in the middle part of the core; however, productivity proxies here are only slightly higher than the bottom of the core, but not high at all in relation to the full record. In addition, not all of the listed proxies show an increase in productivity during this interval (see for example, Ba). It would be more accurate to say that many of these proxies increased during Termination or just after the glacial. Furthermore in this paragraph, not all proxies decrease between 230 and 190 cm.

In several places, but first on line 175 (later, line 186), the authors associate high productivity with warming, but no sea surface temperature proxies are presented in the paper. It is unclear where this association comes from. If it is only from the association of high productivity with climatically warm periods, i.e. Holocene Thermal Maximum, then the sentence on lines 174-176 contains circular reasoning.

On lines 192-196, the authors assert that the percent coarse fraction, magnetic susceptibility, and volcanic glass can be used as a proxy for ice rafted debris, however it is unclear how these were used. Was an index created of the three to track IRD? This should be clarified.

Lines 213-214 indicate that the Tiedemann/Max age model is tuned to the oxygen isotope record from NGRIP, but it is not. In addition, if it was that would cause the main conclusions of this manuscript to employ circular reasoning. This should be clarified so that it is evident that it was a conclusion of Max et al., (2012, 2014) that b* from 12KL correlates to NGRIP even with an independent age model in 12KL.

Section 5.1 is interesting, but seems lengthy and tangential to the discussion at hand. Four pages and two figures are dedicated to reviewing the relationship between paleoclimate in Greenland, Antarctica, and East Asia without any mention of the core that is the subject of the paper. Likely, some of this information is necessary to back up the idea that the Northwest Pacific acted synchronously with East Asia and Greenland, and out of phase with Antarctica, but it needs to be condensed and better organized.

In lines 390-393, the authors state that there is increased sea ice based on their coarse fraction and magnetic susceptibility records, however there is no basis for these claims. Coarse fraction is highest between 18 and 19 ka, not between 15.5 and 17.8 ka. What percent coarse fraction indicates sea ice? I'm not aware of a citation for this from the Pacific or for sea ice specifically, though this is a common indicator of glacial ice rafted debris in the North Atlantic. Also, the coarse fraction presented in Fig. 4 is significantly different from that in Fig. 2. Has Fig. 4 been modified to account for volcanic glass? If so, how was that transformation completed. This should be clearly noted on the figure and in the text.

Technical Corrections

The excessive use of acronyms adds to the reader's difficulty in understanding this manuscript.

Line 20: This should read, "occurred synchronously."

Line 48: Space missing before Max et al., 2012 reference.

Line 88: Please add the word, "the" before "joint Russian-Chinese expedition."

Line 109: color b* "correlates well" not "well correlates."

Line 114: This should read, "from THE 125-250 um fraction."

Lines 182-183: I have no idea what "mechanisms likely similar to established earlier regularities at the orbital-millennial scale" means.

Line 206: I cannot find an explanation of what RPI stands for.

Line 293: what is a "smoothed warmer condition"?

Line 573: Is this the full reference for Harada, 2006?

Table 1: There is no need to include both calendar age in years and calendar age in ka. Note that the date at 306 cm indicates that this foraminifera is 16,016 years old and 14.616 ka old. Which is it?

Figure 2, 3, 4: Please note that the scales for magnetic parameters are reversed.

Figure 3: Could this figure be clarified/condensed in any way? It's a bit overwhelming. In addition, some of the lines of correlation are missing in some cores. Is that intentional?

---

## Author Comment (AC1) · 31 Jan 2017

Anonymous Referee #1 This study presents high-resolution proxy data derived from a marine sediment record of the Northwest Pacific. Based on lithophysical and productivity proxies (e.g. magnetic properties, TOC content or chlorin measurements) the authors provide time series of millennial- to centennial-scale North Pacific climate variability from the Last Glacial Maximum (LGM) into the Early Holocene (EH). Several short-term (centennial scale)

productivity changes/cycles have been identified from 20 to 8 ka and are compared to high-resolution records reflecting changes in the East Asian Monsoon (EAM) system as well as climate variability recorded Greenland and Antarctic ice cores. The authors suggest a close relationship between changes in productivity in the northwestern Pacific, changes in EAM and Greenland/North Atlantic climate variability and further speculate about underlying mechanisms to explain North Atlantic - North Pacific climate synchronicity on centennial timescales.

General comments:

As there is currently quite some debate about the relationship of North Atlantic and North Pacific climate variability during the last glacial termination, the study of Gorbarenko et al. is a timely and relevant research topic. Unfortunately, problems regarding the English (grammar and usage) affect the clarity of the writing and sometimes it is quite difficult to follow authors' interpretations. Moreover, there are several issues that need to be discussed before the manuscript should be considered for publicationin Climate of the Past. Please find below my recommendations to improve the qualityof the manuscript: I have some major complaints about the construction of the age model for core 41-2. The first serious concern is about the use of AMS 14C ages derived from benthic foraminifera (E.pacific and U.parvoc.) to establish age control points for core 41-2. More specifically, a constant offset of 1400 years between planktic and benthicforaminifera has been used to correct benthic ages for reservoir effects according to unpublished? and total regional results? of Max et al. (2014). However, looking into the original publication of Max et al. (2014) reveals that planktic – benthic age differences (BP ages) in core 12KL (used for correlation) varied substantially during the last glacial termination and the Holocene and are not constant through time. The variability in BP ages given in core 12KL is thus in conflict with age assumptions made on benthic foraminifers in this study.

Answer. The age difference between benthic and planktic (B-P) in core 12KL has been reported (Max et al., 2014, Fig. 5) with variations between 950-1400 yr. The data of

the B-P age differences since 15 ka BP has been nearly constant of ∼1400 yr with more pronounced changes in the intermediate water in the Bering and Okhotsk Seas. Bottom water of core 12KL (depth of 2145 m) and core 41-2 (depth of 1924 m) is most likely originated from N Pacific deep water. . Comparing the cooling event in NGRIP at 9.3 ka with a cooling at 9.12 ka in our age model provide the additional confirmation of accepting a B-P age difference about 1400 yr for core 41-2 with very small error (less than 300-400 yr).

RC1. Moreover, there is some confusion about correlation of core41-2 to neighbor core 12KL. Several AMS 14C ages have been transferred from core 12KL to 41-2 to improve the stratigraphy. However, it has been stated by the authors that one age dating from core 12KL was not transferred to 41-2 (16.53 ka at 695 cm in core 12KL) but in the original age model of core 12KL the age in 695 cm is 15.9 ka and not 16.53 ka! I feel that the current age model needs to be revised and further improved.

Answer. We are sorry that there is a typo in our text describing the transfer of age 16.53 ka as a tie point to our core. The original calendar age at depth of 695cm in core 12KL is nearly 15.53 ka (see Table 2 in Max et al. (2012) and Table 2 in Max et al. (2014). But we did not use this depth-age as a tie point in revised age model of core 41-2 (Table 2). Instead we used key time point of core 12KL for depth 706 with age of 16.16 ka (Tiedemann/Max age model 2).

RC1. To improve the age model, it would make sense to e.g. check for correlation to well-dated core SO202-7-6 (Serno et al., 2015). This core has an excellent age model consisting of > 40 AMS 14C ages that, in principle, could be transferred to core 41-2.

Answer. Thanks for reviewer's suggestion. Core SO202-7-6 (Serno et al., 2015) has been well-dated and presented an excellent paper. However, we feel the main obstacle of doing correlation to this core is the time resolution SO202-7-6 core record is much lower than in core 41-2 and 12KL as well. The resolutions of these three cores are ∼220 yr, 30yr and 15yr respectively, which means that the resolution of our core LV41-

2 reported here is nearly 10 times higher core SO202-7-6. In fact we cannot see clearly centennial-millennial environmental cycles in core SO202-7-6 but we are able to see them in cores 41-2 and 12KL.

RC1. A major point that has been made is that the NW-Pacific experienced a sequence of millennial – centennial-scale productivity changes during the last 21ka. The authors further suggest an in-phase behavior between changes East Asian Summer Monsoon and to some extent links to Greenland sub-interstadials. I suggest adding another table, which should give the reader information about the occurrence and length of proposed events.

Answer. Thank you, we will add millennial – centennial-scale productivity changes in the table 3.

RC1. I also recognized that cross-correlation has been done to compare time-series of the EAM and Greenland ice core data. What I'm missing here is crosscorrelation between 41-2 and Greenland and EAM records. I strongly recommend cross-correlation between 41-2 and Greenland and EAM records because this could, from a statistical point of view, strengthen the main conclusion of this study. Ideally, the time window for cross-correlation should be reduced (< 1000 years) according to the centennial-scale variability in 41-2 and other climate records.

Answer.Correlation of the millennial scale climate events between N. Pacific and N Atlantic is very important, you are right. But, we think that so correlation have be estimated between best dated N. Pacific and N Atlantic records. Until now, the best dated records for N Pacific region may be the East Asia monsoon records, which strongly linked with NW Pacific paleoceanography; ones for N. Atlantic may be Greenland $\delta$18O ice core records (NGRIP and GISP2). However, an age model of our core 41-2 constructed both, by AMS 14C data of cores 41-2 and 12KL and correlation of productivity cycles with $\delta$18O records of Chinese speleothems. Both approaches are good matched. Nevertheless, age model of core 41-2 is not enough strong compare to Chinese EAM time scale because it based on it. That is why we prefer to correlate the best dated original records. If we will reduce the time window less than 1000yr, errors in dating of these records will influence on related cross-correlation. That is why we confirm time windows as 1000yr, 2000yr and 3000yr with less influence of age's errors.

Specific comments/Technical corrections:

Abstract:

RC1. Line 17: "The core age model" sounds strange. Better is "The age model of core 41-2 is. . .

Answer. Thank you, we will do this.

RC1. Line 19: Please replace "SO-201-2-12KL" by "SO201-2-12KL"

Answer. Thank you, we will do this.

Introduction:

RC1. Line 38-41: Some references (e.g. Kienast et al., 2001 or Seki et al., 2002) are related to studies in the Northeast Pacific. Accordingly, "NW Pacific" should be replaced by "North Pacific"

Answer. Thank you, we will do this.

RC1. Line 42: A more proper citation would be Kühn et al. (2014)

Answer. We present citation of paper of Kuehn et al. (2014) where Dr. Kühn is coauthor.

RC1. Line 49: "atmospheric teleconnections" Please elaborate a little bit more in the Introduction the mechanisms behind atmospheric coupling.

Answer. Thank you, we do this.

RC1. Line 52: . . .Riethdorf et al. (2013) further suggest. . .

Answer. Thank you, we do this.

RC1. Line 57-61: This sentence is quite long and should be subdivided into two sentences. Though a recent study by Praetorius and Mix(2014) based on multidecadal-resolution foraminiferal oxygen isotope records from the Gulf of Alaska reveals a synchronicity of rapid climate shifts between the N Atlantic/Greenland (NGRIP core record) and the NE Pacific between 15.5 to 11 ka, inverse relationships between the Atlantic/Pacific during the Holocene and Heinrich Event (HE) 1 are suggested, while the short-term variability is either not sufficiently resolved or decoupled.

Answer. Thank you, we do this.

RC1. Line 77: "With our methodologically robust age controls". . .please see major complaint about the age model above. It is probably more accurate to say that the age model (in principle) allows to investigate centennial-millennial scale productivity. .

Answer. We believe that our age model based on the AMS 14C datum and correlation of productivity cycles and the relative paleomagnetic intensity variability between cores 41-2 and 12KL, present robust chronology of cores 41-2 and 12KL that allow us to say about linkages of productivity cycles with sub-interstadials in $\delta18O$ records of the East Asian monsoon. We had strongly improved age model part in revised version.

Material and methods:

RC1. Line 87: Is there any citation (cruise report?) available?

Answer. We have not published cruise report with core 41-2. We have this cruise report only for internal use in our Institute.

RC1. Line 106: Please replace "computing" by "calculating"

Answer. Thank you, we do this.

RC1. Line 109-110: A better beginning of these sentence would be: "It has been shown that variability in color b*. . .."

Answer. Thank you, we do this.

RC1. Line 112-124: Please see comments above regarding the robustness of the age model. Do you use the Intcal09 and Marine09 dataset as stated in Line 123-124 or Marine13 calibration curve as stated in Table 1???

Answer. We correct text: All reservoir age corrected 14C data were converted into calendar age by using Calib Rev 6.0 (Stuiver and Reimer, 1993) with the Marine 13 calibration curve (Reimer et al., 2013).

Results:

RC1. Line 180-183: This sentence is confusing. Please clarify.

Answer.Time resolutions of measured in core 41-2 chlorin content and color b*, TOC, CaCO3 content plus magnetic parameters (PM, MS and RPI), and Ba-bio, Br-bio, Si-bio concentration over the LGM-YD periods are nearly 30 years, 15 years and 60 years respectively.

RC1. Line 200-225: This chapter ("Age model") describes the correlation to neighbor core12KL (please see major comments above) and EAM records. I think the whole chapter should be shifted to the beginning of the Results followed by the paragraph describingother parameter (productivity proxies, CF etc.) derived from sediments. At this point it would make sense to give ages (onset and duration) of the centennial-scale productivity events (e.g. by adding another Table; see main comments above)

Answer. Our age model is based on AMS 14C data of core 41-2 plus projected AMS 14C data of core 12KL on the depth of core 41-2. But projection of AMS 14C data of core 12KL on the depth of core 41-2 are based on correlation of the productivity cycles (color, chlorin, TOC CaCO3) and RPI and PM proxies. That is why we need first to present results of the productivity proxies and then their correlation. OK, we will present additional Table with centennial-scale productivity events.

Discussion:

RC1. Line 243-285: This chapter deals with centennial-scale NW-Pacific productivity events and linkages to EAM records and Greenland ice core data. Thus, I suggest to modify the name of this chapter from "N-S hemispheres climatic linkages of. . .." to "Northern Hemisphere climatic linkages. . .". However, studied records are also compared to EPICA data (from the beginning of line 286) and it would make sense to shift this information to another chapter named e.g. "Southern Hemisphere influence on centennial millennial scale productivity. . .."

Answer. May be you are right. But here we combine the Northern Hemisphere centennial-millennial scale climatic cycles and ones in the southern Hemisphere in one figure and try to show N-S Hemispheres linkages.

RC1. There is also other proxy-data available from core12KL (e.g. SST data; Max et al., 2012 or sea-ice variability data; Méheust et al., 2016), which should be discussed. Accordingly, it could make sense to add SST and/or sea ice variability data from core 12KL to Figure 5, which serve as additional evidence for millennial-scale variability in the NW Pacific.

Answer. We added only productivity cycles of the NW Pacific derived from the cores 41-2 and12KL in the Figure 5. The Fig. 4 with productivity proxies and magnetic properties of cores 41-2 and 12KL is overcrowded and additional records will be problematic for figure reading. But we had added the changes in SST derived from core 12KL (Max et al., 2012) in discussion.

RC1. Line 259: missing citation (e.g. Ruth et al., 2007).

Answer. Thank you, we add this citation

RC1. Line 278-279: "stronger winter monsoon supplies more dust to the loess plateau". Please clarify.

Answer. Using a coupled climate model simulation Sun et al. (2011) investigated the effect of a slow-down of AMOC on the monsoon system and found that a stronger

winter EAM accompanied with a reduction in summer monsoon precipitation over East Asia supplies more dust to the Chinese Loess Plateau.

RC1. Line 355-410: These chapters deal with NW Pacific productivity trends over the LGM-HE1 and the Early Holocene. I suggest moving these chapters to the beginning of the Discussion, followed than by discussion of possible mechanisms to explain productivity events (e.g. one chapters describing links to Northern Hemisphere/Southern Hemisphere).

Answer. The main questions in our MS are the centennial -millennial productivity/climate cycles of the NW Pacific. That is why we mostly focus on these cycles and then on the trends in productivity/climate changes over studied period.

RC1. Line 379-384: How does enhanced intermediate water ventilation promotes more nutrients to the euphotic layer? Most studies in the NW Pacific found substantially reduced nutrients/productivity during HS1 (e.g. Riethdorf et al., 2013). The enhanced formation of intermediate water during HS1 has been related to strong expansion of sea-ice and brine rejection as main processes to intensify mid-depth ventilation (e.g. Rella et al., 2012).

Answer. We explain the enhanced formation of intermediate water during HS1also by other reasons: "The diminished AMOC resulted in a major cooling of the N Atlantic surface water and, most likely, reduced water evaporation in the N Atlantic and therefore Atlantic–Pacific moisture transport. This condition facilitates a reduction of precipitation and hence an overall increase of surface water salinity and decrease of surface stratification in the N Pacific. Moreover, this condition promotes an intensification of the intermediate water ventilation in the N Pacific and also nutrient supply into euphotic layer."

RC1. Line 413-415: Millennial-scale climate events are also well known from the North Atlantic during the Holocene (Bond cycles; Bond et al., 1997). Is there any relationship to millennial-scale events in the North Atlantic?

Answer. Thank you. We input record of Bond et al., 2001 in Fig. 5

RC1. Figure 1: The core locations are hardly visible in this figure. Please modify.

Answer. We modify this figure.

RC1. Figure 3: Please remove all datasets, which are not necessary for age model construction. Please blow up a little bit more the y-axis to help the reader identifying variability in the given records, which is hard to recognize in the current figure.

Answer. We make small changes in this fig. (2). But all proxies are very important for understanding.

RC1. Figure 4: Please see comments to Figure 3.

Answer. We significantly modify fig. 3 and instead of number of proxies for core 41-2 present calculated stack.

RC1. Figure 5: It would make sense to implement one proxy dataset from core 41-2 (e.g. Si-bio) for comparison.

Answer. Thank you. We had implement stack of productivity and color "b" (analog of Si-bio) as proxy dataset from core 41-2 in Fig .5.

RC1. References: Bond, G., W. Showers, M. Cheseby, R. Lotti, P. Almasi, P. deMeno-cal, P. Priore, H. Cullen, I. Hajdas, and G. Bonani (1997), A pervasive millennial-scale cycle in North Atlantic Holocene and glacial climates, Science, 278(5341), 1257-1266. Max, L., L. Lembke-Jene, J. R. Riethdorf, R. Tiedemann, D. Nurnberg, H. Kuhn, and A. Mackensen (2014), Pulses of enhanced North Pacific Intermediate Water ventilation from the Okhotsk Sea and Bering Sea during the last deglaciation, Climate of the Past, 10(2), 591-605. Meheust, M., R. Stein, K. Fahl, L. Max, and J. R. Riethdorf (2016), High-resolution IP25-based reconstruction of sea-ice variability in the western North Pacific and Bering Sea during the past 18,000 years, Geo-Mar Lett, 36(2), 101-111. Rella, S. F., R. Tada, K. Nagashima, M. Ikehara, T. Itaki, K. Ohkushi, T. Sakamoto, N.

Harada, and M. Uchida (2012), Abrupt changes of intermediate water properties on the northeastern slope of the Bering Sea during the last glacial and deglacial period, Paleoceanography, 27. C6 Ruth, U., M. Bigler, R. Rothlisberger, M. L. Siggaard-Andersen, S. Kipfstuhl, K. Goto- Azuma, M. E. Hansson, S. J. Johnsen, H. Y. Lu, and J. P. Steffensen (2007), Ice core evidence for a very tight link between North Atlantic and east Asian glacial climate, Geophysical Research Letters, 34(3). Serno, S., G. Winckler, R. F. Anderson, E. Maier, H. J. Ren, R. Gersonde, and G. H. Haug (2015), Comparing dust flux records from the Subarctic North Pacific and Greenland: Implications for atmospheric transport to Greenland and for the application of dust as a chronostratigraphic tool, Paleoceanography, 30(6), 583-600.

---

## Author Comment (AC2) · 31 Jan 2017

General Comments.

This paper presents a very high-resolution record of productivity in the Northwest Pacific for the past 21 kyrs. It is really a remarkable amount of data especially for the discretely sampled analyses (TOC, grain size, etc). The paper attempts to answer

the question of whether productivity changes in the North Pacific were synchronous or asynchronous from changes in North Atlantic climate. They conclude that increased productivity in the North Pacific is positively correlated with warm intervals in Greenland C1CPD Interactive comment Printer-friendly version Discussion paper and attribute this to "tight atmospheric teleconnections." There are several major issues with this manuscript that make it difficult to follow the authors' conclusions. Specifically, issues with English grammar and word choice make it very difficult to understand at times, the age model causes extreme concern, and the assignment of high productivity/warm events appears to be arbitrary for the core studied (41-2). I highlight a few issues with the word choice and organization of the paper below in "Technical Corrections," however this should not be considered an exhaustive list as I tried to avoid copy-editing as much as possible. It is possible that I have misunderstood how the authors constructed the age model, however it appears that they first radiocarbon dated a mix of benthic and planktonic foraminifera. They then applied a correction of 1400 years to the benthic foraminifera based on the data in Max et al., (2014). They then compared productivity cycles in 41- 2 and 12KL to refine their age model and discarded four of the five radiocarbon ages from core 41-2. Finally, they correlated their productivity cycles to well-dated Chinese subinterstadials. This presents serious concerns for several reasons: RC2: 1. It's unclear why the authors are using benthic foraminifera at all. Are planktonics not plentiful enough? Why weren't radiocarbon samples chosen from the CaCO3 peaks? Surely there should be enough planktic foraminifera when carbonate is high.

Answer. We had used benthic foraminifera because planktonic ones were very rare. There are insignificant carbonate peaks in core 41-2 and carbonate content was low (look Fig. 2).

RC2: 2. It's unclear at which point the authors apply the ventilation conversion– before or after calibrating–after would be appropriate.

Answer. We applied the ventilation conversion after calibrating.

RC2: 3. It's unclear why they chose 1400 as a stable ventilation age. One of the main points of Max et al., 2014 is that ventilation changes dramatically and frequently during deglaciation. In that paper it ranges from 160 to more than 2500 years. It might have been more appropriate to use ventilation estimates corresponding to the approximate C2CPD Interactive comment Printer-friendly version Discussion paper calibrated age.

Answer. The age difference between benthic and planktic (B-P) in core 41-2 for depth 120 cm was accepted as 1400 year according to smoothed spline interpolation for NW Pacific deep water ventilation rate after 15 ka (Max et al., 2014, Fig. 5). Bottom water of core 12KL (water depth of 2145 m) and core 41-2 (depth of 1924 m) is most likely originated from N Pacific deep water. AMS 14C age for depth 120 cm by accepting by a B-P age difference nearly 1400 yr equals to 9.12 ka and separated by our the centennial-millennial low productivity/cold event nearly this interval (Fig. 2) closely correlated with NGRIP cooling event of 9.3 ka.

RC2: 4. It's unclear how the productivity cycles are determined in 41-2. The authors state that they're "based on a suite of productivity proxies and PM records" and that they correspond to synchronous changes in these proxies. However, I have examined several of these intervals and cannot find the commonalities between them. As I understand it, the authors are using Ba, Br, Si, b*, TOC, CaCO3, and chlorin as productivity proxies. Looking first at event 6 from the Last Glacial, I see that some of these proxies are flat during this event, some are fluctuating, and some are high. This pattern continues for all the other productivity events as well. Barium seems to most often be high during productivity events, but there are several peaks in Barium that are not associated with productivity events, why not? Clarifying the determination of these events is essential to all aspects of this manuscript.

Answer. We consider that using suite of productivity proxies instead of 1-3 ones provide more realistic information about productivity changes. Each productivity proxy has own specific response to surface water primary production changes, many papers devote these problem: Dymond et al., 1992 for Barium, Yu et al for CaCO3 content, Sarnthein

et al., for TOC, Gorbarenko et al., 2014 for Si-bio (color "b") et so on. It is subject for special paper. Different productivity proxies respond to other kinds of productivity effect on content of TOC, chlorin; preservation in sediment of separated proxies, etc. The presentation of the suite of productivity proxies is our advantage. Beside productivity proxies we also used the PM record because the sediment PM reflects the changes in transportation of dust from continent by atmosphere circulation associated with climate changes. Yes, not all proxies change synchronously but their common trends allow us to trace productivity events. Therefore we used term quasi-synchronous. For statistic we calculated a productivity stack of all productivity proxies and reversed PM index to show more clearly (statistically estimated) occurrence of the centennial-millennial productivity events in the NW Pacific which was presented in revised Figs. 3 and 5. Therefore we did some changes in related paragraph of manuscript (lines . . .).

RC2: 5. It's unclear how the authors determined which radiocarbon ages to discard. They discarded three benthic ages and the only planktic age measured. The planktic age is probably the strongest part of the age model and it seems to fit in with their correlation to 12KL. Why is it discarded? Why is one benthic age kept, but not others?

Answer. We kept benthic age for depth 120 cm because it is younger age for core 41-2 and this data also good matched with projected age of core 12KL for depth 126 cm. Laying below AMS data of core 41-2 are good matched with projected AMS 14C datum from core 2KL and confirm validity of projection 14C data core 12KL on depth of core 41-2. We do not discard them and show them in Figs 2 and 3. But sediment of core 41-2 has lower carbonate content and do not characterized by such significant CaCO3 peaks as in core 12KL. That is why we prefer to use 14C data projected from core 12 KL in our final age model for core 41-2, because 14C data for core 12 KL were determined for planktonic foram from depth with strong Ca peaks.

RC2: 6. The final correlation of the productivity events to Chinese subinterstadials is perhaps the most troubling part of this age model. If the age model for 41-2 is tuned to the oxygen isotopes from Chinese caves, then the authors can not claim

that productivity events in the North Pacific happened synchronously with these sub-interstadials. This is circular logic. In addition, it would be useful to include a discussion that addresses the differences between the myriad productivity proxies. By no means do these records look the same, especially on centennial to millennial scales. Why not?

Answer. In context of critical comments of reviewers 1 and 2 we try more clearly present age model of core 41-2 construction. We had yet explained using of suite productivity proxies in identification of the productivity events in point 4. Then we show how we find that centennial-millennial productivity events in NW Pacific core had occurred synchronously with summer EAM sub Interstadial: "A close time correlation of these NW Pacific productivity increasings/environmental amelioration events with sub-interstadials in summer EAM become apparent after projection of the radiocarbon datum of both cores on the absolute U-Th dated $\delta$18O record of the China caves stalagmites (Wang et al., 2008; ) over the 20-8 ka BP (Fig. 3)", lines . . ..

Specific Comments

RC2: Section 2.1: It's unclear how tephra was estimated. Was it identified under a microscope? By magnetic susceptibility, some kind of Principal Component Analysis of the XRF scanning? I'm not sure what "semi-quantitative component analysis of this fraction with a total of 12 ranged scales" means.

Answer. In the section of 2 Materials and methods we tell that content of the terrigenous particles, tephra, planktonic and benthic forams and other component in the course sediment fraction (CF) was semi -quantitatively estimated under binocular with a total of 12 ranged scales. We used the comparative percentage charts for estimating proportions of sedimentary components (Rothwell, 1989).These results allow us to separate tephra input from IRD one (only terrigenous particles) in the measured CF values (Fig. 2).

RC2: When I first read Section 2.4, I was under the impression that this manuscript

would present planktic-benthic pairs of radiocarbon dated foraminifera. I needed to examine the table myself to determine that it does not. This should be clarified.

Answer. Table 1 show AMS 14C measurements of the benthic and planktic foraminifera picked in several depth intervals and their calibration.

RC2: In Section 2.6, it's unclear how the terrigenous component of Ba, Br, and Si was determined. Lines 159-160 read, "The terrigenous component, in turn, was calculated from empirical regional (Ba/Al)ter ratios in the sediment core with the lowest Ba-tot contents." What sediment core are you referring to? Where is it located? What is the regional observed value (Ba/Al)ter? If this is an empirical value, it should not be vague.

Answer. The empirical (Ba/Al)ter ratio was estimated for the studied core 41-2 using the technique suggested by Goldberg et al. (2005). The exact value of the ratio was calculated separately for each of data series.

RC2: The age model should come before any discussion of sediment ages in the Results Section, i.e. Sections 3 and 4 should be reversed (and revised accordingly).

Answer. Our age model of core 41-2 was based on the correlation of the productivity events between the cores 41-2 with ones of well dated core 12KL. That is why we need put results with productivity proxies and separated productivity events before an age model section.

RC2: In line 169, the authors posit that they observe high productivity in the middle part of the core; however, productivity proxies here are only slightly higher than the bottom of the core, but not high at all in relation to the full record. In addition, not all of the listed proxies show an increase in productivity during this interval (see for example, Ba). It would be more accurate to say that many of these proxies increased during Termination or just after the glacial. Furthermore in this paragraph, not all proxies decrease between 230 and 190 cm.

Answer. ALL PRODUCTIVITY PROXIES demonstrate higher productivity in interval

of 230-315 cm, as was stated in line 169, including the Ba-bio as well (Fig. 2). This interval was correlated with BA warming by us, consistently with available for this core AMS 14C data and found earlier for studied region high productivity during BA warming. ALL PRODUCTIVITY PROXIES demonstrate also productivity decrease in interval of 190-230 cm, correlated by our with YD cooling, and following productivity increase from 190 cm was associated with Holocene warming that is also consistent with known regional climate and productivity trend and AMS 14C data.

RC2: In several places, but first on line 175 (later, line 186), the authors associate high productivity with warming, but no sea surface temperature proxies are presented in the paper. It is unclear where this association comes from. If it is only from the association of high productivity with climatically warm periods, i.e. Holocene Thermal Maximum, then the sentence on lines 174-176 contains circular reasoning.

Answer. We had clarify the connection of productivity changes with environment and climate changes on the millennial scale in revised ms and discussed SST changes versus time for core 12 KL (Max et al., 2012).

RC2: On lines 192-196, the authors assert that the percent coarse fraction, magnetic susceptibility, and volcanic glass can be used as a proxy for ice rafted debris, however it is unclear how these were used. Was an index created of the three to track IRD? This should be clarified.

Answer. We had clarified this in revised version of paragraph 2.1 (Coarse fraction measurement). We show that CF and MS records may be used as IRD only with controlling of tephra share input in the CF. When the tephra share is large in CF (Fig. 2), we can't determine IRD input.

RC2: Lines 213-214 indicate that the Tiedemann/Max age model is tuned to the oxygen isotope record from NGRIP, but it is not. In addition, if it was that would cause the main conclusions of this manuscript to employ circular reasoning. This should be clarified so that it is evident that it was a conclusion of Max et al., (2012, 2014) that b* from 12KL

correlates to NGRIP even with an independent age model in 12KL.

Answer. Thank you for this comment. We had significantly revised section related to the age model and show how we used AMS 14C datum of core 12KL and correlation of color "b" with NGRIP $\delta$18O curve according to Max et al. (2012, 2014), lines .

RC2: Section 5.1 is interesting, but seems lengthy and tangential to the discussion at hand. Four pages and two figures are dedicated to reviewing the relationship between paleoclimate in Greenland, Antarctica, and East Asia without any mention of the core that is the subject of the paper. Likely, some of this information is necessary to back up the idea that the Northwest Pacific acted synchronously with East Asia and Greenland, and out of phase with Antarctica, but it needs to be condensed and better organized.

Answer. It is important comment. We added the productivity proxy records in Fig. 5 in order to test centennial-millennial productivity events in the NW Pacific. Here we present statistically significant stack of productivity for core 41-2. Causal relationship of identified centennial-millennial productivity events in the NW Pacific (yellow bars) with paleoclimate changes in the East Asia, the Greenland and Antarctica and solar irradiance variability for the last 14 kyr was discussed in this section. The productivity records of cores 41-2 and 12KL with the shown productivity events were discussed in Sections 5.1 and 5.2.

RC2: In lines 390-393, the authors state that there is increased sea ice based on their coarse fraction and magnetic susceptibility records, however there is no basis for these claims. Coarse fraction is highest between 18 and 19 ka, not between 15.5 and 17.8 ka. What percent coarse fraction indicates sea ice? I'm not aware of a citation for this from the Pacific or for sea ice specifically, though this is a common indicator of glacial ice rafted debris in the North Atlantic. Also, the coarse fraction presented in Fig. 4 is significantly different from that in Fig. 2. Has Fig. 4 been modified to account for volcanic glass? If so, how was that transformation completed. This should be clearly noted on the figure and in the text.

Answer. We show trends of CF and MS records in Fig. 4 which reflect IRD trend input in the sediments off Kamchatka up to nearly 10 ka, when tephra share in CF significantly rise and discussed this in revised text. Records of CF and MS are similar in the Figs. 2 and 4; only in Fig. 4 they were presented versus age.

Technical Corrections

RC2: The excessive use of acronyms adds to the reader's difficulty in understanding this manuscript. Line 20: This should read, "occurred synchronously." Line 48: Space missing before Max et al., 2012 reference. Line 88: Please add the word, "the" before "joint Russian-Chinese expedition." Line 109: color b* "correlates well" not "well correlates." Line 114: This should read, "from THE 125-250 um fraction." Lines 182-183: I have no idea what "mechanisms likely similar to established earlier regularities at the orbital-millennial scale" means.

Answer: We add some references in order to show "established earlier regularities" in the revised text and discussed the linkages of the sharp productivity events with climate and environmental changes in the N Pacific and its marginal seas.

RC2: Line 206: I cannot find an explanation of what RPI stands for.

Answer: "Synchronous pattern of RPI variability in the far NW Pacific variability" means that relative paleointensity of magnetic field of Earth change synchronously in the past. Therefore RPI curves, recorded of sediment cores 41-2 and 12KL have to change synchronously versus time and may be used for time correlation.

RC2: Line 293: what is a "smoothed warmer condition"?

Answer: Thank you. We change this sentence in revised text: warming in the Antarctica at 23.6-24.4 kyr was . . ..

RC2: Line 573: Is this the full reference for Harada, 2006?

Answer: We modified this reference. This cruise report is available on-line.

RC2: Table 1: There is no need to include both calendar age in years and calendar age in ka. Note that the date at 306 cm indicates that this foraminifera is 16,016 years old and 14.616 ka old. Which is it?

Answer: Thank you. We delete the column with calendar ages in year. Calendar age of sediment at depth of 306 cm equal to 14.61 ka.

RC2: Figure 2, 3, 4: Please note that the scales for magnetic parameters are reversed.

Answer: Thank you. We note this in revised Fig captures for PM.

RC2: Figure 3: Could this figure be clarified/condensed in any way? It's a bit overwhelming.

Answer. We condense Fig. 3 by showing the productivity stack.

RC2: In addition, some of the lines of correlation are missing in some cores. Is that intentional?

---

## Author Comment (AC3) · 31 Jan 2017

[revised manuscript text omitted]
 as for interstadials events in the NW Pacific and Okhotsk and Bering Seas. The rises in

SST of surface water and environment amelioration in the NW Pacific and Japan, Okhotsk and

Bering Seas correlated with interstadials in $\delta^{18}O$ records in NGRIP ice core (North Greenland Ice

Core Project members, 2004) and Chinese cave stalagmites promote to increase in productivity at the millennial scales (Gorbarenko et al., 2005; Nagashima et al., 2011; Seki et al., 2002,

2004). Although an each used productivity proxy have own specific peculiarities in his response to climate and environmental changes, the used complex of proxies allow us to more definitely determine increased productivity events in the past. In results, presented productivity proxies and sediment paramagnetic magnetization (PM) records show that 6 short increased productivity/warmer events happened during the last glacial and 4 ones during the B/A warming (Fig. 2). During the EH we find 5 short lower productivity/colder events and 3 higher productivity/warmer events. We notice that a colder event at depth 117-122 cm with an age of

~9.12 ka (Table 1) is well-correlated with the 9.3 ka cold event in Greenland ice core records (Rasmussen et al., 2014). Moreover, a colder event identified at depth 106-109 cm of core 41-2

also links well with the 8.2 ka cold event in Greenland ice cores, a well-known chronostratigraphic marker in the Early to Middle Holocene boundary (Walker et al., 2012).

The share of tephra in sediment CF show significantly increase in upper part of core since

130 cm (Fig. 2); therefore, below this interval, CF and MS variability was mostly responded to

IRD input. The CF and MS records, controlled by tephra share in CF, indicate high IRD input in sediment of lower part of core and strong decrease to top in interval 325-315 cm. MS and CF

records also show some increase of IRD at the interval of 230-200 cm related to the YD (Fig. 2).

Available productivity proxies (chlorin, Ca/Ti ratio, color b*) plus magnetic properties RPI

and PM for core 12KL were compared with results of core 41-2 versus cores depth (Fig. 3). For simplicity, a suite of productivity proxies for core 41-2 (color b* and chlorin, TOC, CaCO3, Ba- bio, Br-bio and Si-bio content and PM record) was replaced with the calculated stack of productivity proxies. The color b* index and Ca/Ti ratios (analog of $CaCO_3$ content) of core

12KL were extracted from Max et al. (2012, 2014) available on PANGAEA Data Publisher for

Earth & Environmental Science (http://dx.doi.org/10.1594/PANGAEA.830222).

**4. Age model**

An age model of core 41-2 was constructed using all available AMS [14]C dating, with more age control points identified by correlating the centennial-millennial events of the productivity proxies, RPI and PM of studied core with those of the well-dated nearby core 12KL (Max et al.,

2012, 2014) (Fig. 3). The age tuning used in this study assumes a synchronous pattern of productivity, RPI and PM variability in the far NW Pacific since the last glacial especially for close located cores. With this conception of age model developments, the centennial-millennial variability of productivity proxies with increased productivity events, relative paleointensity (RPI) of Earth magnetic field and paramagnetic magnetization (PM) identified in cores 41-2 and

12KL have to be closely matched in the both cores over the last glaciation, the B/A warming to the EH (Fig. 3). We noticed that the available for core 12KL the Tiedemann/Max age model 2

(Max et al., 2012, 2014) was based on the AMS $^{14}$C data and correlation of color b* index with the NGRIP $\delta^{18}$O curve (PANGAEA Data Publisher). By adopting an age model of core 41-2,

AMS $^{14}$C dating of core 12KL of Max et al. (2012, 2014) were transferred successfully to core

41-2 according to correlation of related increased productivity events and RPI values (Fig. 3).

Color b* minimum in core 12KL at depth of 706 cm, which R. Tiedemann/L. Max correlates with minimum in NGRIP $\delta^{18}$O curve at 16.16 ka, is also clearly correlate with color b* minimum of core 41-2 at depth of 348 cm (Fig. 3). All correlated AMS $^{14}$C key points are also well- matched between measured RPI curves of both cores (Fig. 3). Core 41-2 AMS $^{14}$C data of 9.45

ka, 10.6 ka , 14.39 ka and 14.61 ka at depth 127.5 cm, 156 cm, 298 cm and 306 cm respectively are rather closed to nearby projected $^{14}$C datum from core 12 KL (Table 2) and confirm validity of this age projection. But here we prefer to used $^{14}$C data of core 12KL because this core have higher sedimentation rate and planktonic foraminifera for these measurements were picked from intervals with highest Ca content that significant decrease a bioturbation effect.

A close time correlation of these NW Pacific productivity increasing/environmental amelioration events with sub-interstadials in summer EAM become apparent after projection of the radiocarbon datum of both cores on the absolute U-Th dated $\delta^{18}$O record of the China caves stalagmites (Wang et al., 2008) over the 20-8 ka (Fig. 3). Such inferred synchronicity of NE

Pacific productivity abrupt events and EAM sub-interstadials was used for further fine age model construction by tuning of increased productivity events with related sub-interstadials of $\delta^{18}$O

Chinese stalagmites for depth beyond the projected AMS $^{14}$C data (Fig. 3; Table 2).

**5. Discussion**

With the constructed age model of core 41-2 different kinds of productivity proxies and magnetic results combined with some of them for AMS $^{14}$C dated core 12KL (Max et al., 2012,

2014) reveal sequence of noticeable centennial-millennial scale productivity cycles in the far

NW Pacific occurred in-phase with Chinese sub-interstadials (CsI) associated with stronger summer EAM (Wang et al., 2008) over the 21–8 ka (Fig. 4). These linkages suggest the centennial-millennial increase productivity events in the far NW Pacific were likely associated with shifts to warmer climate and/or higher nutrient conditions in surface water synchronously with CsI of the summer EAM. Presented high resolution records show clearly that three centennial-millennial increase productivity/environment amelioration events correlated with (CsI) had occurred during the LGM, three CsIs during the HE1, four CsIs during the B/A

warming, and three CsIs during the EH (Fig. 4) (Table 3). The possible mechanisms responsible for the in-phase relationships or the synchronicity of the centennial-millennial scale events between the NW Pacific productivity and summer EAM are discussed and proposed below.

**5.1. N-S hemispheres climatic linkages of centennial-millennial climate/environment**

**changes over the LGM - HE 1- B/A warming**

Identifying any linkages of centennial–millennial climate changes in the Northern

Hemisphere between the NW Pacific, EAM, and N Atlantic/Greenland and the climate changes recorded in Antarctic ice core responsible for the Southern Hemisphere is important to us, to deepen understanding of the mechanisms responsible for the timing and spatial propagation patterns that resulted from the abrupt variability in the global climate and environmental system.

In order to test the linkages, we demonstrate here the correlation among the highly resolved U-

Th dated $\delta^{18}O$ records of the composite Hulu and Dongge caves sediments (Dykoski et al., 2005;

Wang et al., 2008), the ~20-year averaged resolution $\delta^{18}O$ and $Ca^{2+}$ content records of the GISP2

and NGRIP with 5 point running mean on the annual-layer counted GICC05 age scale (Rasmussen et al., 2014), the $\delta^{18}O$ record of the EPICA Dronning Maud Land (EDML) ice core from Antarctica (EPICA Community Members, 2006) on the methane synchronized timescale with the NGRIP core, and the Siberian climate calculated from pollen results of the Lake Baikal region (Bezrukova et al., 2011) over the past 25 ka (Fig. 5). The $Ca^{2+}$ content in the Greenland ice cores serves as a proxy for dust mobilization on the land and transferring in the high latitudes of the N Hemisphere by atmosphere governed by climate and atmosphere circulation changes (Sun et al., 2012). It has been suggested that the nearly synchronous ice core $\delta^{18}$O and $Ca^{2+}$

millennial scale changes reflect the shifting of Greenland atmospheric dust loading which is closely linked with the atmospheric circulation and climate changes in the high-latitude of N

Hemisphere, where EAM plays important role (Ruth et al., 2007). Initially the persistent millennial scale changes shown in the Greenland ice core records were defined as interstadials (GI) and stadial (GS) (Johnsen et al., 1992), but have been refined by INTIMATE stratigraphy studies which introduced the subdivision of the GI-1 into sub-interstadials GI-1a to GI-1e.

Furthermore, the GS-2.1 was subdivided into sub-Stadial GS-2.1a (over the HE 1), GS-2.1b (LGM), and GS-2.1c (Björck et al., 1998; Rasmussen et al., 2014) (Fig. 5).

Established in the studied off Kamchatka cores the NW Pacific centennial-millennial productivity/environment cycles with stack of productivity and color b* on the base of its chronology (Table 2) were put on the N-S hemispheres climate variability over the LGM-HE1-

B/A (Fig. 5). In addition to established in the NW Pacific studied cores the six centennial- millennial productivity/environment cycles over the LGM-HE1, we suggest that coeval with

CsIs an additional three abrupt events likely took place in the NW Pacific over the interval of 25-

20 ka (Fig. 5). Therefore, we found the three EAM/NW Pacific sub-interstadials which occurred within GS-2.1a (namely CsI-GS2.1-1, CsI-GS2.1-2 and CsI-GS2.1-3), four CsIs within GS-2.1b (CsI-GS2.1-4 to CsI-GS2.1-7), and two within GS-2.1c (CsI-GS2.1-8 and CsI-GS2.1-9) (Fig. 5).

It also has been noted that the some $\delta^{18}$O differences in coeval $\delta^{18}$O values between

Summit and NGRIP ice cores over the LGM-HE1 period, were likely governed by changes in the N American Ice Sheet volume and N Atlantic sea-ice extent, that results in the changes of meridional gradients in the Greenland ice $\delta^{18}$O (Seierstad et al., 2014). Probably, such $\delta^{18}$O

differences in the Summit - NGRIP $\delta^{18}$O values may explain that correlation of the EAM/NW

Pacific sub-interstadials with Greenland sub-interstadials recorded in $\delta^{18}$O and $Ca^{2+}$ of the GISP2

and of the NGRIP over the HE1 (GS-2.1a) was less pronounce then ones during LGM. On the basis of the high-resolution NGRIP core investigation (less one year) over 15-11 ka, Steffensen et al. (2008) have suggested that at the beginning of the GI, the initial northern shift of the

Intertropical Convergence Zone (ITCZ), identified in a sharp decrease of dust within a 1-3 year interval, triggered an abrupt shift of Northern Hemisphere atmospheric circulation. Such circulation pattern changes forced a more gradual change (over 50 years) of the Greenland air temperature associated with high latitude atmosphere circulation and westerly jets ways reorganization. Evidence from a loess grain size record in NW Chinese Loess Plateau (Sun et al.,

2012), infer the linkage of the changes in EAM strength and Greenland temperature over the past

60 ka, and suggests a common force driving both changes (Sun et al., 2012). Using a coupled climate model simulation Sun et al. (2011) investigated the effect of a slow-down of AMOC on the monsoon system and found that a stronger winter EAM accompanied with a reduction in summer monsoon precipitation over East Asia supplies more dust to the Chinese Loess Plateau and likely into the NW Pacific. This study indicates that AMOC is a driver of abrupt change in

EAM system, with the northern westerlies as the transmitting mechanism from the N Atlantic to the Asian monsoon regions. Other evidences of teleconnection between the EAM and N Atlantic on a millennial timescale come from the investigation of the Japan Sea sediments. Nagashima et al. (2011) infer that temporal changes in the provenance of eolian dust in Japan Sea sediments reflect changes in the westerly jet path over East Asia, happened in-phase with Dansgaard-

Oeschger cycles.

EPICA community members (2006) show that methane synchronization of the EDML and the NGRIP $\delta^{18}$O records reveal one-to-one alignment of each Antarctic warming with a corresponding stadial in Greenland ice cores, implying a mechanism of bipolar seesaw on these time scales. Changes in the heat and freshwater flux were connected to the AMOC and a stronger

AMOC leads to increased transport of heat from the Southern Ocean heat reservoir. In results of

EAM investigations (Wang et al., 2001) have suggested that between 11,000 and 30,000 yr BP

the Chinese interstadials (CI) recorded in $\delta^{18}$O calcite of cave stalagmites had happened apparently synchronously with the GIs and therefore CIs were, likely, related to Antarctica cold events also. For example, smoothed warmer condition in the Antarctic at 23.6-24.3ka was synchronous with abrupt climate cooling and increases in dust content in the Greenland ice cores

NGRIP and GISP2, coeval to HE2 of the N Atlantic and in-phase with summer EAM weakening (GS/CS-3.1) (Fig. 5). Subsequent Antarctica cooling since 23.4 ka was accompanied by

Greenland warming with two sharp interstadials GI-2.2 and GI-2.1 with nomenclature of

Rasmussen et al. (2014) and China interstadial CI-2 coeval with sub-interstadial CsI-GS2.1-9

associated with summer EAM intensification (Fig. 5). Over the LGM-HE1 period, the most of sub-interstadials in the N hemisphere had occurred during abrupt Antarctica temperature decrease as well (Fig. 5).

It also has been suggested that a monsoon intensity index including the EAM was controlled not only by Northern Hemisphere temperature ('pull' on the monsoon, which is more intense during boreal warm periods), but also by the pole-to-equator temperature gradient in the

Southern Hemisphere ('push' on the monsoon which is more intense during the boreal cold periods) that leads to enhanced boreal summer monsoon intensity and its northward propagation (Rohling et al., 2009; Rossignol-Strick, 1985; Xue et al., 2004). Since the summer EAM

transports heat and moisture from the West Pacific Warm Pool (WPWP) across the equator and to higher northern latitudes (Wang et al., 2001), the temperature gradient in the Southern

Hemisphere "pushes" the summer EAM intensity by means of its influence on the latitudinal/longitudinal migrations or expansion/contraction of the WPWP. This also explains the difference of responses of EAM and Greenland interstadials and sub-interstadials, because the migration of the WPWP may have responded more slowly than the atmospheric changes. All the above interpretations are mostly consistent with variability between the EAM and Antarctica temperature (Fig. 5), when cooling in Antarctica promote to increase summer EAM. The $\delta^{18}$O

record of Chinese EAM changes were more gradual then in the $\delta^{18}$O of Greenland ice cores, and the amplitude changes of the EAM are more similar to the Antarctic air temperature changes (Fig. 5).

During B/A warming when Antarctic temperature was decreased, four EAM sub- interstadials (CsI-GI1-a – CsI-GI1-e) coeval with established NW Pacific centennial-millennial productivity/environment cycles have varied in-phase with Greenland sub-interstadials (Björck et al., 1998) as well (Fig. 5). Recent high-resolution investigations on Bering Sea sediment cores from the "Bering Green Belt" (Kuehn et al., 2014) have documented four well-dated laminated sediment layers during the B/A warming-beginning of Holocene, with three of them within the

[revised manuscript text omitted]

2014) by moving windows at 1000, 2000 and 3000 years show their more significant synchronization (correlation ranging from -0.6 to -0.9) during period of 16.5-8.5 ka (Fig. 6).

During earlier (25-16.5 ka) and later (8.5-1 ka) periods there are differences in CC between the

EAM-NGRIP and the EAM-GISP2. But both CC during these periods demonstrate occurrence of weak synchronization/or absence of significant correlation (within ranging at ±0.25) (Fig. 6).

Significant synchronization was indicated also by CC between EAM-NGRIP during the Middle-

Late Holocene. More discrepancies in both CC were observed over 19.5-16.5 ka, that may be explained by errors in ages measurements and/or by different atmospheric teleconnection between the EAM and the GISP2/NGRIP cores due to its differ locations in the Greenland. So the statistics imply that the seesaw mechanism between the EAM/NW Pacific and the

Greenland/N Atlantic during 25-1 ka is not effective, however being in line with empiric results of the EAM and the Greenland teleconnection by shifting of the westerly jet path (Nagashima et al., 2011; Sun et al., 2012).

**5.3 NW Pacific productivity trends over the LGM-HE 1**

Beside of the centennial -millennial productivity/environmental cycles, we find common

NW Pacific productivity trends over the LGM and HE1 with some differences in other types of productivity proxies. According to the sharp increase of Antarctica temperature, dust content in the Greenland ice cores and significant decrease in the summer EAM we put boundary of

LGM/HE1 on nearly the 17.8 ka (Fig. 5). This age is a little earlier to that being placed at ~17.5

ka which is a timing for the beginning of catastrophic iceberg discharges in the HE1, but nearly coincides with the abrupt increase of the $^{231}$Pa/$^{230}$Th ratio in the N Atlantic core OCE326-GGC5, which marks the beginning of the collapse of AMOC (McManus et al., 2004).

During the LGM most of productivity proxies demonstrate minimum primary production in the far NW Pacific without definite trends, although the Si-bio of core 41-2 and color b* of core 12KL show even small negative trend through time (Fig. 4). Severe environmental condition in the central Asia inferred from vegetation reconstruction (Bezrukova et al., 2011)

(Fig. 5) promote to increase in winter sea ice covering consistently with high IRD accumulation in studied region inferred from CF and MS records (Fig. 4) that hamper productivity. It is in concord with early established minimum of productivity in the NW Pacific due to strong stratification prevented nutrients supply for supporting productivity in surface waters (Gebhardt et al., 2008).

Since 17.8 up to 15.3 ka, the core 41-2 productivity proxies such as Ba-bio, Br-bio, TOC

and chlorin associated with production of calcareous phytoplankton (mostly coccolithophores), show significant increased trends simultaneous to gradual Antarctic warming accompanied by strongly diminished of AMOC (McManus et al., 2004). The diminished AMOC resulted in a major cooling of the N Atlantic surface water and, most likely, reduced water evaporation in the

N Atlantic and therefore Atlantic–Pacific moisture transport. This condition facilitates a reduction of precipitation and hence an overall increase of surface water salinity and decrease of surface stratification in the N Pacific. Moreover, this condition promotes an intensification of the intermediate water ventilation in the N Pacific and therefore nutrient supply into euphotic layer.

The observed trends of productivity proxies are in concord with strong intensification of the intermediate depth water ventilation in the N Pacific during HE1 (Max et al., 2014) based on the

$\delta^{13}C$ foraminifera data from the intermediate water and radiocarbon-derived ventilation ages.

However, rather constant $CaCO_3$ values in both cores (water depth 1924-2145 m) during LGM-

HE1 do not indicate the changes of the water ventilation at these depths in the N Pacific over that time span because carbonate concentration in the sediment strongly defined by the ventilation (Yu et al., 2014). While the productivity proxies Si-bio and color b*, associated with siliceous phytoplankton production (mostly diatoms), do not show significant trends since HE1 to ~15.3

ka. The strong sea ice effect with high IRD input up to 15.3 ka, shown by CF and MS records, (Figs. 2; 4) was more significant in our studied area and likely overwhelm the productions of diatom algae for coccolithophores due to a large spring-early summer surface water stratification during seasonal sea ice melting.

A sharp increase of NW Pacific primary production and rise of diatom production since

~15.3 ka indicated by most productivity proxies and Si-bio and color b* records with culmination at sub-interstadial GI1-e of B/A warming was, likely, induced by decrease of sea ice influence and its spring melting favoring for weakening of surface stratification (Figs. 4 and 2).

The timing of decrease in the sea ice cover since ~15.3 ka is consistent with the surface water warming (Max et al., 2012) and with the central Asia vegetation/environment amelioration inferred by Bezrukova et al. (2011) by pollen reconstructions (Fig. 5). Such pattern of the productivity changes in the N Pacific and the Bering Sea during glacial/interglacial transition was observed in other cores (Caissie et al., 2010; Galbraith et al., 2007; Gebhardt et al., 2008;

Keigwin, 1998) and was, likely, a persistent feature for the N Pacific and its realm forced by resumption of the AMOC at the B/A beginning coeval with the cooling in the Antarctica (Fig. 5).

In the Okhotsk Sea, being strongly intruded in the NE Asia continent, the beginning of the diatom production and accumulation of the diatomaceous sediments had occurred only since the

Middle Holocene (5-6 kyr BP) due to the later diminish of sea-ice cover and later breakdown of spring/early summer surface water stratification (Gorbarenko et al., 2014).

**6. Conclusion**

This study presents high resolution records of a suite of productivity proxies (TOC,

CaCO$_3$, chlorin, color b*, Ba-bio, Br-bio, Si-bio), sediment lithological (CF) and magnetic properties (PM, MS and RPI) from a sediment core 41-2 taken from the NW Pacific (East

Kamchatka slope). Presented results reveal a sequence of 13 centennial-millennial scale regional productivity increased/environment amelioration events over the LGM-EH (20-8 ka) in the far

NW Pacific.

The age model of core 41-2 was constructed by using available AMS [14]C dating, with more age control points identified by correlating the centennial-millennial events of the productivity proxies, RPI and PM of studied core with those of the well-dated nearby core 12KL

(Max et al., 2012, 2014). Thus all available AMS [14]C dating of core 12KL were transferred successfully to core 41-2. Based on projected radiocarbon datum of both cores on the $\delta^{18}$O

record of the Chinese caves stalagmites (Wang et al., 2008) the close time correlation of NW

Pacific productivity events with sub-interstadials in summer EAM over the 20-8 ka was used for further fine age model construction. In results, established 
[revised manuscript text omitted]

---

## Author Response (AR1)

**Editor Decision: Reconsider after major revisions** (10 Feb 2017) by Dr. Erin McClymont
Comments to the Author:
Both reviewers have commented favourably on the value of the very detailed, multi-proxy and high-resolution study provided in this publication. It is clear that there could be some very valuable new information revealed by the data, but at present the relevance and support for those linkages is masked by a number of factors: (1) a lack of clarity around several components of the age control; (2) a lack of clarity on how the productivity proxies are interpreted, and what processes drive each of these; (3) difficulties in following the discussion where links are made between the East Asian Monsoon, Greenland, and Antarctica.

I agree with the reviewers, and think that they make a compelling set of arguments for the corrections required for the manuscript text. Due to the density of data being presented here, it is also likely that very detailed graphics will result. But, as the reviewers also pointed out, some of these become overwhelming to negotiate. The authors have replied to the comments provided by the reviewers, in places clarifying their interpretations and/or methods. However, I still found several components of the discussion to be confusing, or not clarified to the extent that was sought by the reviewers. I have outlined below what is required to make a convincing case that a robust set of arguments is put forward in this manuscript, which will require more clarification that the information currently given in the authors replies to the reviewers. As a result, I recommend that the manuscript is reconsidered only after major revisions are made.

1. Age model construction
I agree with both reviewers that there are some serious concerns around the construction of the age model and how it is outlined in the main text. It is difficult to follow exactly what was done, where points were transferred or not, why certain offsets were considered, and how the tuning of productivity cycles was tested. The replies given to the reviewers are not totally convincing: a revised submission needs to very carefully lead the reader through each step, and justify the approach.

Answer. The age model construction of core 41-2 is not straight-forward but step-wisely based on millennial to centennial scales of productivity and climate in the NW Pacific.

First, in core 41-2 with high sedimentation and high resolution sampling we have documented the centennial millennial productivity/climate events in the NW Pacific and only numbered them (without any timing) (Fig. 2).

Second, having orbital scale stratigraphy and one AMS 14 C data of core 41-2 with age of 9.12ka, we applied graphic correlation of productivity/climate events and PRI curve of this core with ones of the well-dated nearby core 12KL. In frame of this correlation we projected AMS 14C data of core 12KL on the depth of core 41-2, while taking into account the RPI variability of both cores as well. Projected AMS 14C data from core 12KL that were measured on the planktic foraminifera and age model 2 of Tiedemann/Max of this core is a valid approach (PANGAEA Data Publisher) as the adjacent locations of two cores.   Other four AMS 14C data of core 41-2 were used only to re-confirm the validity provided by the AMS data projection (Fig. 3) on core 41-2 depth and have not been used explicitly in our age model.

Third, with the initial controls of age model from the two steps described in above, we put established AMS 14C data of both cores on the coeval ages of precisely dated δ18O record of the Hulu cave stalagmites. In results, it was able to see, that centennial millennial productivity/climate events in the NW Pacific cores had occurred in-phased with sub-interstadial of summer EAM system with a convincible structural similarity of the events (Fig. 3). This similarity allows us to construct our age model in more detailed by tuning of the all centennial millennial productivity/climate events in NW Pacific cores with relative dated sub-interstadial of summer EAM. In revised manuscript we have described the construction of our age model step by step. As the construction of our age model is not that straight forward, in the manuscript we thus had to present the section with productivity events and then an age model construction.

There is a discussion of benthic-planktonic offsets and whether they vary through time – the authors reply that above 15 ka this offset was constant in the core from which they transfer their age model. But there are fluctuations shown in the Max et al. (2014) paper which also become pronounced before 15 ka – and this new manuscript includes older samples which might have been affected by these intervals. Furthermore, if there are large changes in productivity (which might reflect changes in mixing and/or carbon export) could these not also have affected these planktonic-benthic offsets? Answer.Very approximately interpolation of the regional deep water ventilation age of Max et al. (2014) show that deep water ventilation was nearly constant before 15 ka (nearly 1400 year) and vary between 1400-2000 year in an interval 15-19 kyr. We had used only younger AMS 14C data of core 41-2 for our age model with accepted BF-PF age difference nearly 1400 year, which age (9.2 ka) marks low productivity/cold event closed to established in the Greenland cold event 9.3 ka. As we responded earlier, we did not use other four radiocarbon data for core 41-2 in our final age model, though they fit quite closely with framework of core 12KL age model.

A revised submission needs to more carefully outline the steps taken to generate the age model and the assumptions which have gone into it. There is confusion about where tuning to Greenland and/or East Asian monsoon records might have occurred: this needs to be clarified. I agree with the reviewers that the construction of the age model would be better placed before discussion the productivity records, because then the text can flow into the timings of the productivity changes not just the overall pattern. This could be achieved if there was an outline of what the different productivity proxies represent and so we might expect that e.g. b* can be used to tune between sites. At the moment, the text talks about the timings of changes and cycles in the productivity data BEFORE the age control has been justified.

2. Productivity proxies and their interpretation
A number of different proxies for productivity are presented. The paper does not review the benefits and limitations of these different proxies, to make a clear statement of how they should be interpreted. This means that, where the reviewer comments that different proxies do different things at different times, it is then difficult to know which record to accept, and when, and why. A link needs to be made at the site first between what these proxies mean, and then how the oscillations are interpreted. The reviewers may have been able to understand the tuning approaches and the descriptions of productivity trends if there was a stronger rationale of the data interpretation. In a revised submission I recommend that the interpretations of the proxies are outlined with more detail so that the reader knows how you are finding a productivity argument (e.g. preservation can affect several of your indicators but is not discussed).

Answer. Usually people present in their results only 1-3 productivity proxies and suggest that they exactly reflect productivity changes without discussion of any their limitations and preferential. We agreed overwhelming with the review comments and therefore added a brief review of productivity proxies being used in this study and also a discussion on their limitations. It is worthy to emphasize in our manuscript as the centennial millennial productivity events are key issues of our study.

A. In Method section, we added brief implication of each used method, especially for the productivity proxies.
B. In Result section, we described briefly limitation and peculiarity of productivity proxies. That is why presentation a wide range of productivity proxies allows us to take into account different aspects transformation of primary produced organic matter into different proxies and their preservation in sediment providing more realistic picture of productivity changes. We provide graphic correlation of all used productivity proxies in order to establish the centennial millennial productivity events in studied core in the Fig. 2. In Fig. 3, we provide the productivity stack instead of used productivity proxies; its variations are consistent with graphic correlation. The stack reduced amount of curves on the figure and understanding of figure become easier.

I am not totally convinced by the use of the 'stack', but that is because without a revised manuscript to see what you did it is not clear to me what this represents. Is this an average of normalised data from several proxies? Is it a weighted mean? And is this even an appropriate way to create a kind of summary of the data, when you note in the text that sometimes the proxies behave in different ways? Given the complexity of the data, did the authors try any form of additional statistical analysis to determine whether there are 'groups' of productivity indicators or patterns?
In a revised submission you will need to very clearly explain what went into the stack, why those parameters were chosen, whether there is susceptibility in the stack to one or two of the records etc.
Answer. It's well-known that the productivity proxies, used in this study, initially are controlled by surface water productivity, but the productivity proxies are also susceptible to controlling factors other than productivity, such as preservation or early diagenesis. In the revised text we added a description that, in order to find out a commonly joint pattern of the multiple productivity proxies, we presented a productivity stack calculated from an average normalized data in range of 0-1 for all used productivity proxies plus reversed paramagnetic magnetization (PM) because this parameter reflects climate changes. We had used an equal weight for all proxies in the stacking calculation as it is the best solution for encompassing all available proxies without any preferential for any particular proxies before the full mechanisms of biological or geochemical processes that have controlled the proxies are fully understood.

3. Teleconnections
Reviewer 1 makes valuable comments about the way that the text links the monsoon and Greenland records and/or with Antarctica. I agree with their comment that a slight restructuring or separating of the sections across lines 243-285 would be valuable, to first set the NW Pacific data into a hemispheric context, and only then discuss links with the south. This doesn't have to be a problem for the figure (as the authors suggest): more than one section of the text can cite the figure.

Answer. Thank you. We revised this piece of text significant and had discussed first set the NW Pacific data into N hemispheric and then into S hemisphere.

I also agree with reviewer 1 that explaining the longer term trends first, in your interpretation, would allow you to move from the long term into the short term variability. The structure might then flow better for the comparisons with wider teleconnections and their timescales.

Answer. Main issue of our ms, as reflected in the title, is centennial-millennial productivity/climate events in the NW Pacific and their teleconnection with sub-interstadials in EAM system and then with Greenland and Antarctic temperature abrupt changes. So, we mainly focused on describing our finding of the time scales.Longer term trend interpretation of our results is of course important but we have placed it as secondary. Sorry, therefore we prefer to save previous order of presentation.

I also agree with reviewer 1 that when making the cross-correlations for teleconnections (see below), what would make a stronger discussion of the significance of your data is a cross-correlation of this new data with the monsoon/Greenland records (whichever is appropriate).

Answer. In this chapter, we try to correlate the N Pacific climate changes with ones on the N Atlantic. Our age model based on AMS 14C data and fine-tuning with well dated $\delta^{18}O$ of China stalagmites reflected the on EAM temporal activity which, likely, are the best dated available template for NW Pacific region. Therefore, the age model of our data is a derivative of the EAM chronologies for the first approximation. That is why we used EAM data as a best approach of NW Pacific for cross-correlation with the NGRIP and GISP2 data.

First, find the mechanisms and links with your own data, and then (possibly) make the bigger hemisphere/polar comparisons.

Answer. We briefly show links of NW Pacific abrupt climatic/productivity events with EAM millennial variability in text. Bigger hemisphere/polar comparison was discussed and presented in Fig.5.

4. Graphics
A positive aspect of this publication is the number of detailed new data sets, showing multiple indicators of productivity at high temporal resolution. The reviewers made some suggestions for how to simplify these – the use of a 'stack' in a revised figure 3 was one response by the authors. I agree with the reviewers that the graphics are still complex. One way to try and simplify e.g. figure 3, might be to plot the data sets that you are tuning side-by-side e.g. plot both RPI records together, then both b* together, or whatever the combination might be. Or in fact was it the 'stack' that was used for the tuning.

Answer.On the figure 3 proxy curves are plotted versus their own depth, so we cannot put side-by-side records from different cores together due to different sedimentation rates in these cores. For tuning we use whole bunch of productivity proxies and PM. These records reflect environmental changes more or less definite depend on time period. The obtained stack combines all features of all used records. The main purpose of this stack curve is to simplify our figures and clearly figure out the centennial-millennial productivity events.

The revised figure 4 is more clear.

Figure 5 is a bit strange/messy with simply an addition of the Bond 2001 hematite grains overlying Antarctica.

Answer. We changed the order of curves on the figure 5

I'm not convinced that Figure 6 is needed, unless you include (as one reviewer suggests) the cross-correlation of your own data into this. Discussion of monsoon/Greenland climate phasing is an important question, but to better embed this into your new findings you need to include a component of your data here.

Answer. We presented figure 6 to show different correlation (connection) between NW Pacific and Greenland/N Atlantic in various time intervals. As we mentioned above our age model is based on EAM chronologies as the best available dating template for NWPacific region. Therefore, the age model of our data is partly converted from the EAM chronologies and that is why we used it for cross-correlation with the NGRIP and GISP2 data.

Or, remove the detailed discussion of the wider teleconnections from the manuscript and focus more strongly on your new data and its relationships to e.g. monsoon, Greenland, Antarctica.
Answer. We are not sure what exactly you meant here, because the teleconnection and relations to monsoon, Greenland, Antarctica are the same issue in our opinion.

The reviewers noted that there is other data on e.g. SST, sea ice, available which could help you to make the links between the mechanisms you outline to change productivity and the potential data which could support these. In your reply to reviewers you say that you will state the relationships: I don't think that this is what the reviewers are asking for. I, like them, would prefer to see you plot the SST and/or sea ice evidence alongside your own.
Answer. SST estimation technique is not well developed for the high latitude oceans. Alkenons concentration strongly vary with time here, therefore estimations of SST are always challenging works here. With the published SST results for core 12Kl presented by Max et al. (2012, Climate of the Past) and Meyer and Max, et al. (2016, Paleoceanography), SST in the first paper (Max et al., 2012) was determined by $U^K_{37}$ method and show low SST during YD. SST in the second paper (Meyer, Max et al., 2016) for the same core was determined by TEX86 method and showed that during the YD, the SSTs have significantly decreased and abruptly increased. The patterns have been found difficult to interpret as far.
All climate results for NW Pacific and its marginal Seas, adjoin land and in other regions presented in many papers evident that YD was cold period with increase of sea ice extension and low productivity. With such modelling and data results we are quite confident that the YD must be cooling nearly everywhere in the NW Pacific and East Asia, though SST determinations for the ocean responses during the YD possibly are more complicated due to various effects that are unclear.

As for sea ice evidence. We have shown that the presented MS and CF may be used as IRD proxy. They show that IRD content (sea ice influence index) was high during LGM-HE 1, then significant decrease from 15.3 ka ago and increase during YD. We had presented many proxies reflected environment and productivity changes and find centennial-millennial productivity/climate events in the NW Pacific coeval with EAM sub-interstadials. We believe that it is important.

5. Written style
Both reviewers comment that there are grammatical errors, leading to confusion and/or a lack of clarity in the arguments and presentation of data. I agree with this comment. A thorough and careful reading of the manuscript is required to remove these errors, to ensure that your arguments are being made with improved precision.

Answer. We used the proofreading service from native English speaker.

[revised manuscript text omitted]

---

## Author Response (AR2)

Editor Decision: Reconsider after major revisions (15 May 2017) by Erin McClymont

Comments to the Author (pdf): cp-2016-102-comments-to-author.pdf

Comments to the Author:

The authors have replied to both reviewers, and submitted a revised version of the text. The main issues raised by the reviewers and the editor centred on:

1. Age model construction. This is generally improved in the revised submission, but the authors still resist moving the construction of the age model before discussion of the proxies. I agree that it is difficult to identify the correct ordering here, but since there is no discussion of what the productivity records mean before they are tuned (this comes later), the structure of the text is very difficult to follow. If the authors wish to keep the age model after the discussion of the productivity proxies and what they might mean, so that there is some process-based rationale for why you might expect different records to align with other sites and with the monsoon. This is particularly key when one considers that the different proxies don't always show the same patterns, so that a reason why some peaks were tuned and not others needs to be stated. A revised submission needs to either (1) place the age model construction first, before discussion of the productivity signals; (2) explain first what the proxy signals record, to make the link to why tuning could or should be appropriate (see next point)

Answer 1. In revised text we first present additional information how the different seven productivity proxies were responded for primary productivity changes and how centennial events with higher productivity were separated and numbered through studied core. Here we also explained why different variability of productivity proxies may be occurred not everywhere synchronicity. So, the more remarkable peaks in detrended stack of productivity present more reliable NW Pacific centennial productivity events.

Only then, we discussed an age model construction and show why we conclude that NW Pacific centennial productivity events are correlated in time with EA summer monsoon sub interstadials.

2. Productivity proxies and their interpretation. The authors do now include some additional information on how the productivity proxies could and should be used, but this needs to come earlier in the discussion, to justify why these different proxies were used and how they have been interpreted. This is particularly important for the generation of the stack, and what it means. I don't agree with the authors that this is the subject of a different paper: you are using these data to make bold statements about NW Pacific productivity changes, but as Reviewer 2 points out, the selection of which peaks are being used seems to vary through time and be quite arbitrary. It needs justification. This would also address the concerns of point 3 below, where the focus on the NW Pacific would be a better demonstration of the value of the next records, but give detail in how we should interpret them, especially where they may disagree or show variable responses.

Answer 2. We had revised text according these comments and comments of Editor in PDF file" cp-2016-102-comments-to-author. 15.05.17"and answers for them. This PDF file with author answer are applicated.

3. Teleconnections. I still agree with reviewer 1 that you should undertake the cross-correlations for your own, new data, rather than shifting to an analysis of previously published records of monsoon variability. Yes, you have tuned your data to the monsoon data, but what would be most interesting here is to learn whether the productivity in the NW Pacific Ocean could be linked to millennial scale events elsewhere (see also comment by Reviewer 2 that section 5.1 is largely tangential – I agree).

You note that not all of the same proxies show the same patterns: would it not be interesting to detail those relationships, which may evolve through time and give some interesting insights into what controls NW Pacific Ocean circulation and productivity? There must be factors beyond monsoon intensity which affect the regional primary productivity, which could be interesting to learn.

Answer. Follow to reviewer 1, we present cross-correlation of the productivity stack with δ18O of NGRIP record In the revised version.

We strongly revised former section 5.1 and more focus on the different mechanisms which controlled the NW Pacific centennial productivity events during LGM-HE1, B/A and EH periods.

Here is where the SST data could be very useful (Reviewer 1), rather than the repeated assumption throughout the revised text that productive equals warmth (cold oceans can be productive, depending upon the driver).

Answer. We present references for SST data of core 12 KL and other cores and show that events with increased productivity in the NW Pacific had happened during abrupt climate warming.

Reviewer 2 also raises concern about the circular logic of tuning to the monsoon and then describing the synchroneity of the events between the two regions.

Answer. We infer the synchronicity of the NW Pacific centennial events with increased productivity with sub interstadials of EA summer monsoon after projection the radiocarbon datum of both cores with productivity events on the absolute U-Th dated $\delta^{18}$O record of Chinese stalagmites.

The focus on the NW Pacific data and its variability has not been addressed in the revised manuscript, and I continue to be concerned that the main data analysis and discussion in the document is not about the new data which has been presented here. A revised version needs to address this concern by focussing more on the new data and less on the monsoon-Greenland teleconnections.

Answer. We closely follow these comments in revised MS

4. Graphics. These are better in the revised manuscript.

5. Written style. Whilst some of the corrections made have clarified the text, there remain areas of grammatical errors which require careful editing. I have highlighted a number of these on the tracked-changes document, but there is work to be done here.

Non-public comments to the Author:

Dear authors,

I apologies for the delays in replying to your submitted documents. I hope you appreciate that given the complexities of the detailed reviews we received, and the responses that you made, that it was important that full consideration was given as to whether your revision addressed the concerns raised.

Whilst efforts have been made to address many of the comments, I still have concern about the structuring of the current version, particularly in terms of where the discussion of the productivity proxies sits.

I am also very concerned that you continue to focus on the links between EAM (not your data) and Greenland/Antarctica (also not your data) in your discussion. You have presented some very interesting and detailed data for the NW Pacific, yet your discussion focusses instead on the importance and teleconnections of the records which were your tuning targets. I continue to agree with reviewer 2 that it would be better to focus on your new findings from the NW Pacific, perhaps considering what that data might contribute to discussions of teleconnections. You have provided some explanation of what might drive the productivity cycles when you outline the principles of what drives those records, but as I note above there must be other (interesting?) processes beyond the monsoon which can explain productivity variations, especially since your proxies do not always match in their directions and amplitudes. At the moment the significance and richness of your new data is diminished by a discussion focussed on data which is already published and not your own.

Answer. Thank you very much for these comments. We try to fully understand your concerning's and respectively do revision of text.

I have made many comments on your revised submission, and attached it here. Please take the time to read these from start to finish, because you will find some positive statements about text which may simply need to be moved and restructured to address some of the concerns.

If you feel that you are able to address the concerns still noted in my public comments above, and the edits and questions I have made to your manuscript, I would be happy to consider reviewing a revised submission.

Answer. We look carefully all your comments in the former revised manuscript and give answer for them. They very help us to improve structure and text of MS. Thank you a lot again.

Best wishes

Erin McClymont

Marked-up manuscript version

**Centennial to millennial climate variability in the far northwestern Pacific (off Kamchatka) and its linkage to the East Asian monsoon and North Atlantic from the Last Glacial Maximum to the Early Holocene**

Sergey A. Gorbarenko [1], Xuefa Shi [2, 3], Min-Te Chen [3], Mikhail I. Malakhov†, [4], Galina Yu. Malakhova, [5],

Aleksandr A. Bosin [1], Yanguang Liu [2, 3], Jianjun Zou [2, 3]

[1] V.I. Il'ichev Pacific Oceanological Institute, Russia

[2] Key Laboratory of Marine Sedimentology and Environmental Geology, First Institute of Oceanography, SOA, Qingdao, China

[3][3] Laboratory for Marine Geology, Qingdao National Laboratory for Marine Science and Technology, Qingdao, China

[4] National Taiwan Ocean University

[4[5] North-East Interdisciplinary Science Research Institute FEB RAS, Russia

**Abstract**

High resolution reconstructions based on productivity proxies and magnetic properties measured from sedimentof core LV63-41-2 (off Kamchatka),) reveal prevailing centennial - millennial productivity/climate variability in the northwestern (NW) Pacific from the Last Glacial Maximum (LGM) to the Early Holocene (EH). The age model of the core 41-2 is established by AMS $^{14}$C dating using foraminifera shells and by correlating the projections of AMS $^{14}$C data of the nearby core SO-201-12KL through correlation of the productivity cyclesproxies and relative paleomagnetic intensity records with those of well dated nearby core SO201 12KL. Our results show a pronounced feature. Resulted sequence of centennial - millennial productivity increases/climate cycles ofwarming events in the NW Pacific had occurred synchronicitysynchronously with the summer East Asian Summer Monsoon (EAM) at

[revised manuscript text omitted]
 as for interstadials events in the NW Pacific and Okhotsk and Bering Seas. The rises in SST of surface water and environment amelioration in the NW Pacific and Japan, Okhotsk and Bering Seas correlated with interstadials in $\delta^{18}O$ records in NGRIP ice core (North Greenland Ice Core Project members, 2004) and Chinese cave stalagmites promote to increase in productivity at the millennial scales (Gorbarenko et al., 2005; Nagashima et al., 2011; Seki et al., 2002, 2004). Although an each used productivity proxy have own specific peculiarities in his response to climate and environmental changes, the used complex of proxies allow us to more definitely determine increased productivity events in the past. In results, presented productivity proxies and sediment paramagnetic magnetization (PM) records show that 6 short increased productivity/warmer events happened during the last glacial and 4 ones during the B/A warming (Fig. 2). During the EH we find 5 short lower productivity/colder events and 3 higher productivity/warmer events. We notice that a colder event at depth 117-122 cm with an age of ~9.12 ka (Table 1) is well-correlated with the 9.3 ka cold event in Greenland ice core records (Rasmussen et al., 2014). Moreover, a colder event identified at depth 106-109 cm of core 41-2 also links well with the 8.2 ka cold event in Greenland ice cores, a well-known chronostratigraphic marker in the Early to Middle Holocene boundary (Walker et al., 2012).

The share of tephra in sediment CF show significantly increase in upper part of core since 130 cm (Fig. 2); therefore, below this interval, CF and MS variability was mostly responded to IRD input. The CF and MS records, controlled by tephra share in CF, indicate high IRD input in sediment of lower part of core and strong decrease to top in interval 325-315 cm. MS and CF records also show some increase of IRD at the interval of 230-200 cm related to the YD (Fig. 2).

Available productivity proxies (chlorin, Ca/Ti ratio, color b*) plus magnetic properties RPI and PM for core 12KL were compared with results of core 41-2 versus cores depth (Fig. 3). For simplicity, a suite of productivity proxies for core 41-2 (color b* and chlorin, TOC, CaCO3, Ba-bio, Br-bio and Si-bio content and PM record) was replaced with the calculated stack of productivity proxies. The color b* index and Ca/Ti ratios (analog of $CaCO_3$ content) of core 12KL were extracted from Max et al. (2012, 2014) available on PANGAEA Data Publisher for Earth & Environmental Science (http://dx.doi.org/10.1594/PANGAEA.830222).

**4. Age model**

An age model of core 41-2 was constructed using all available AMS [14]C dating, with more age control points identified by correlating the centennial-millennial events of the productivity proxies, RPI and PM of studied core with those of the well-dated nearby core 12KL (Max et al., 2012, 2014) (Fig. 3). The age tuning used in this study assumes a synchronous pattern of productivity, RPI and PM variability in the far NW Pacific since the last glacial especially for close located cores. With this conception of age model developments, the centennial-millennial variability of productivity proxies with increased productivity events, relative paleointensity (RPI) of Earth magnetic field and paramagnetic magnetization (PM) identified in cores 41-2 and 12KL have to be closely matched in the both cores over the last glaciation, the B/A warming to the EH (Fig. 3). We noticed that the available for core 12KL the Tiedemann/Max age model 2 (Max et al., 2012, 2014) was based on the AMS [14]C data and correlation of color b* index with the NGRIP $\delta^{18}O$ curve (PANGAEA Data Publisher). By adopting an age model of core 41-2,

AMS $^{14}$C dating of core 12KL of Max et al. (2012, 2014) were transferred successfully to core 41-2 according to correlation of related increased productivity events and RPI values (Fig. 3). Color b* minimum in core 12KL at depth of 706 cm, which R. Tiedemann/L. Max correlates with minimum in NGRIP $\delta^{18}$O curve at 16.16 ka, is also clearly correlate with color b* minimum of core 41-2 at depth of 348 cm (Fig. 3). All correlated AMS $^{14}$C key points are also well-matched between measured RPI curves of both cores (Fig. 3). Core 41-2 AMS $^{14}$C data of 9.45 ka, 10.6 ka , 14.39 ka and 14.61 ka at depth 127.5 cm, 156 cm, 298 cm and 306 cm respectively are rather closed to nearby projected $^{14}$C datum from core 12 KL (Table 2) and confirm validity of this age projection. But here we prefer to used $^{14}$C data of core 12KL because this core have 
[revised manuscript text omitted]

With the constructed age model of core 41-2 different kinds of productivity proxies and magnetic results combined with some of them for AMS [14]C dated core 12KL (Max et al., 2012, 2014) reveal sequence of noticeable centennial-millennial scale productivity cycles in the far NW Pacific occurred in-phase with Chinese sub-interstadials (CsI) associated with stronger summer EAM (Wang et al., 2008) over the 21–8 ka (Fig. 4). These linkages suggest the centennial-millennial increase productivity events in the far NW Pacific were likely associated with shifts to warmer climate and/or higher nutrient conditions in surface water synchronously with CsI of the summer EAM. Presented high resolution records show clearly that three centennial-millennial increase productivity/environment amelioration events correlated with (CsI) had occurred during the LGM, three CsIs during the HE1, four CsIs during the B/A warming, and three CsIs during the EH (Fig. 4) (Table 3). The possible mechanisms responsible for the in-phase relationships or the synchronicity of the centennial-millennial scale events between the NW Pacific productivity and summer EAM are discussed and proposed below.

[revised manuscript text omitted]

associated with summer EAM intensification (Fig. 5). Over the LGM-HE1 period, the most of sub-interstadials in the N hemisphere had occurred during abrupt Antarctica temperature decrease as well (Fig. 5).

It also has been suggested that a monsoon intensity index including the EAM was controlled not only by Northern Hemisphere temperature ('pull' on the monsoon, which is more intense during boreal warm periods), but also by the pole-to-equator temperature gradient in the Southern Hemisphere ('push' on the monsoon which is more intense during the boreal cold periods) that leads to enhanced boreal summer monsoon intensity and its northward propagation (Rohling et al., 2009; Rossignol-Strick, 1985; Xue et al., 2004). Since the summer EAM transports heat and moisture from the West Pacific Warm Pool (WPWP) across the equator and to higher northern latitudes (Wang et al., 2001), the temperature gradient in the Southern Hemisphere "pushes" the summer EAM intensity by means of its influence on the latitudinal/longitudinal migrations or expansion/contraction of the WPWP. This also explains the difference of responses of EAM and Greenland interstadials and sub-interstadials, because the migration of the WPWP may have responded more slowly than the atmospheric changes. All the above interpretations are mostly consistent with variability between the EAM and Antarctica temperature (Fig. 5), when cooling in Antarctica promote to increase summer EAM. The $\delta^{18}$O record of Chinese EAM changes were more gradual then in the $\delta^{18}$O of Greenland ice cores, and the amplitude changes of the EAM are more similar to the Antarctic air temperature changes (Fig. 5).

During B/A warming when Antarctic temperature was decreased, four EAM sub-interstadials (CsI GI1-a – CsI GI1-e) coeval with established NW Pacific centennial-millennial productivity/environment cycles have varied in phase with Greenland sub-interstadials (Björck et al., 1998) as well (Fig. 5). Recent high-resolution investigations on Bering Sea sediment cores from the "Bering Green Belt" (Kuehn et al., 2014) have documented four well-dated laminated sediment layers during the B/A warming-beginning of Holocene, with three of them within the

[revised manuscript text omitted]

---

## Author Response (AR3)

Dear Erin McClymont,

Thank you very much for your comments. We rectified all noted grammatical corrections. Also we change order of authors and improved the Fig. 1..